# A selective and atom-economic rearrangement of uridine by cascade biocatalysis for production of pseudouridine

Martin Pfeiffer[1,2], Andrej Ribar[1,2] & Bernd Nidetzky [1,2] ✉

As a crucial factor of their therapeutic efficacy, the currently marketed mRNA vaccines feature uniform substitution of uridine (**U**) by the corresponding *C*-nucleoside, pseudouridine (**Ψ**), in 1-N-methylated form. Synthetic supply of the mRNA building block (1-N-Me-**Ψ**−5′-triphosphate) involves expedient access to **Ψ** as the principal challenge. Here, we show selective and atom-economic 1*N*-5*C* rearrangement of β-D-ribosyl on uracil to obtain **Ψ** from unprotected **U** in quantitative yield. One-pot cascade transformation of **U** in four enzyme-catalyzed steps, via D-ribose (Rib)-1-phosphate, Rib-5-phosphate (Rib5P) and **Ψ**-5′-phosphate (**Ψ**MP), gives **Ψ**. Coordinated function of the coupled enzymes in the overall rearrangement necessitates specific release of phosphate from the **Ψ**MP, but not from the intermediary ribose phosphates. Discovery of Yjjg as **Ψ**MP-specific phosphatase enables internally controlled regeneration of phosphate as catalytic reagent. With driving force provided from the net *N-C* rearrangement, the optimized **U** reaction yields a supersaturated product solution (∼250 g/L) from which the pure **Ψ** crystallizes (90% recovery). Scale up to 25 g isolated product at enzyme turnovers of ∼$10^5$ mol/mol demonstrates a robust process technology, promising for **Ψ** production. Our study identifies a multistep rearrangement reaction, realized by cascade biocatalysis, for *C*-nucleoside synthesis in high efficiency.

β-Pseudouridine (**Ψ**, **1a**) is the *C*-nucleoside isomer of the canonical nucleoside uridine (**U**, **2**). **Ψ** occurs naturally in all forms of RNA[1–3]. **U** replacement by **Ψ** generally enhances RNA stability, typically by effect on RNA secondary structure[4]. **Ψ** has played a key role in technology development for mRNA vaccines[5]. Early works reveal that synthetic mRNAs are less immunogenic and are translated more efficiently in vivo when **U** is uniformly substituted by **Ψ**[6,7]. Flexibility of the RNA polymerase enables **Ψ** to be incorporated via in vitro transcription in the presence of **Ψ**-5′-triphosphate (**Ψ**TP, **3a**) instead of UTP[8,9]. Later studies show that the 1-N-methyl derivative (N1m**Ψ**, **1b**) is even more potent than **Ψ** in decreasing synthetic mRNA immunogenicity[10]. The N1m**Ψ**, therefore, replaces **U** in the current mRNA-based COVID-19 vaccines and is essential to their therapeutic efficacy[5,11,12]. Up to now

(02.2023), 13.0 billion doses of these vaccines have been administered to humans worldwide. In addition, more than 60 new mRNA vaccines for the treatment of cancer and infectious diseases are in clinical trials at the moment[11]. Besides vaccines, **Ψ**-modified mRNA is promising to boost the gene-cutting efficiency in CRISPR-based gene editing[13,14]. Overall, therefore, a rapidly growing demand for the RNA building blocks **Ψ**TP and N1m**Ψ**TP (**3b**) is envisioned for the near future.

Synthetic supply of the *C*-nucleoside triphosphates involves chemistries developed over decades of research (Fig. 1b). Here, expedient access to the **Ψ** β-*C*-glycoside core structure represents the principal challenge. Methylation at N1 and conversion into the 5′-triphosphates proceed by transformations well implemented for process scale use[15–19]. Since the first chemical synthesis of **Ψ** in 1961[20], the synthetic

[1]Institute of Biotechnology and Biochemical Engineering, Graz University of Technology, NAWI Graz, Petersgasse 12, A-8010 Graz, Austria. [2]Austrian Centre of Industrial Biotechnology (acib), Krenngasse 37, A-8010 Graz, Austria. ✉e-mail: bernd.nidetzky@tugraz.at

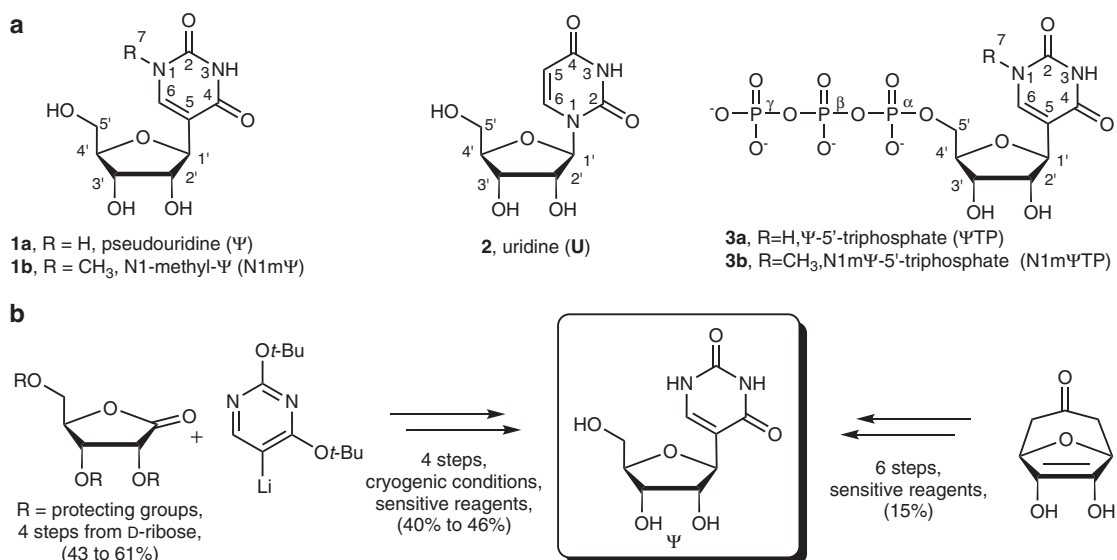

**Fig. 1 | Examples of chemical syntheses of Ψ. a** The *C*-nucleosides Ψ and N1mΨ compared to the *N*-nucleoside isomer **U**. Structures of ΨTP and N1mΨTP as the *C*-nucleotides used for therapeutic mRNA production. **b** Exemplary synthesis routes towards Ψ based on direct coupling of the nucleobase and ribonolactone[29,30] or the reconstruction of the uracil nucleobase on a ribofuranosyl precursor[25].

routes explored in particular detail (Fig. 1b)[21–24] have been the reconstruction of the uracil nucleobase on a C1′-funcionalized D-ribofuranosyl moiety[25–28] or the coupling of a suitably primed heteroaryl with a glycosyl reagent, such as glycal or pentonolactone[18,20,29–33]. Despite considerable efforts in method development, stereo-direction (β:α) remains an important issue[29,30,33–35]. Starting from D-ribonic acid 1,4-lactone in a most advanced synthesis, Ψ is obtained in 40% yield over three steps (Fig. 1b). Besides limitations due to low reaction efficiency, the chemical methods of Ψ synthesis also raise concern of sustainability and economic viability of their use in production. The requirement for cryogenic reaction conditions, protecting group chemistry and the additional usage of non-benign chemicals detract severely from process eco-friendliness.

Biocatalysis offers an attractive alternative to pure chemical synthesis, overcoming many of the mentioned drawbacks[36–43]. In *N*-nucleoside synthesis, enzymatic routes are well known[43–47] and the recent examples of Molnupiravir[48–50] and Islatravir[51] showcase the potential of multienzyme cascade reactions for process scale production[48,51–54]. The biosynthesis of Ψ happens at **U** sites within RNA, through 1*N*–5*C* rearrangement catalyzed by Ψ synthase[55–57]. The free **U** or its nucleotides (UMP, UDP, UTP) are not used by the enzyme[58], presenting a roadblock to an atom-economic transformation (**U** → Ψ) that could be considered a kind of organic "dream reaction" for the synthesis of Ψ. Despite the growing number of *C*-riboside-forming enzymes discovered recently from natural pathways[59–72], no candidate catalyst of a plausible synthetic route to Ψ emerges.

We therefore here focused on the Ψ-5′-monophosphate (ΨMP) *C*-glycosidase YeiN[68,69,73,74]. The enzyme catalyzes naturally the hydrolysis of ΨMP into uracil (Ura) and D-ribose 5-phosphate (Rib5P)[69]. The reverse YeiN reaction was promising for Ψ synthesis because of two characteristics in particular[73]. Unlikely for a nominal hydrolysis performed in water, the reaction equilibrium lies far on the ΨMP side; and YeiN provides absolute β-stereo-direction of the *C*-ribosylation of uracil[69,73]. Reaction of the isolated ΨMP with an unspecific phosphatase gives Ψ[73,74]. Asking whether the YeiN reaction could become suitable for Ψ production in principle, we immediately noticed the fundamental need for reaction integration through an innovative (cascade) reaction concept. The chemically intriguing *N*-*C* rearrangement by Ψ synthase on RNA-bound **U** drew our attention. This led to the idea of a multienzyme cascade transformation, with

*C*-glycosylation as the central step, to achieve the same net conversion of the free **U**. The **U** is an expedient starting material available in bulk quantities by large-scale fermentation[75–77]. Inspired by the natural salvage pathways of pyrimidine nucleosides, we here combine the YeiN (EC 4.2.1.70) reaction with the reactions of **U** phosphorylase (EC 2.4.2.3) and phosphopentomutase (EC 5.4.2.7) to promote a one-pot transformation of **U** into ΨMP (Fig. 2).

The further addition of a ΨMP-specific phosphatase (EC 3.1.3.5), discovered in this study for the purpose of controlled recycling of free phosphate, enables the **U** rearrangement into Ψ (Fig. 2). The **U** transformations into ΨMP and Ψ are both diastereoselective and atom-economic. Driven by the β-*C*-glycosylation, they proceed in quantitative yield to final concentrations at the product solubility limit and with a productivity of ~40 g/L/h. Performed at a scale of 25 g isolated Ψ, the enzyme cascade reaction is demonstrated as a robust biocatalytic process technology that is highly promising for use in production.

## Results

### Design of the multienzyme cascade reaction

Domino reaction in three or four enzymatic steps was designed for **U** conversion into ΨMP or Ψ, respectively (Fig. 2). As shown in Table 1, the phosphorolysis of **U** involves only a limited driving force to give Rib1P and uracil, but the isomerization of Rib1P into Rib5P advances thermodynamically downhill. There is additional pull from the C-C coupling so that the overall conversion of **U** into ΨMP should readily proceed. The ΨMP hydrolysis is largely irreversible. The chemical reactions do not involve an immediate proton uptake or release. Phosphate group protonation does not change over the different steps (Rib1P, Rib5P, ΨMP). pH independence of the overall reaction equilibrium is thus implied, eliminating the pH as a factor of the chemical transformation. All reactants are stable under mild reaction conditions in water. Reactant solubility is discussed later.

Given the requirement of up to four enzymatic reactions telescoped in one pot, we considered enzyme compatibility regarding dependence of their activity on the bulk conditions (e.g., pH, temperature, ionic strength). The YeiN used is from *E. coli* and so we selected the already known other enzymes (**U** phosphorylase, UP; phosphopentomutase, DeoB) also from *E. coli*. A phosphatase suitable for use in the cascade reaction needed to be discovered. As described in a separate section below, we focused our search on the diversity of

**Fig. 2 | Proposed enzymatic N1-C5 rearrangement from U to Ψ.** The rearrangement proceeds in four enzymatic steps in a one-pot transformation. The enzymes used are identified by name and EC number. Phosphate is recycled in the process.

*E. coli* phosphatases and identified the enzyme Yjjg. Good functional overexpression in *E. coli* was an additional point considered and it was confirmed by the experiments (Table 1, Supplementary Fig. 5). All enzymes were isolated by His-tag chromatography and exhibited specific activities (Table 1) within the range given in the BRENDA database[78]. A holistic approach of multienzyme system characterization (see later) was used to identify operational conditions of pH (7.0) and temperature (30 or 40 °C) for synthesis performed in one-pot transformations. Kinetic parameters of the individual enzymes were determined under these conditions (Supplementary Fig. 1) and are summarized in Table 1. The substrate $K_M$ values are all in the low mM range or even smaller. The substrate concentrations used, and the intermediate concentrations accumulating, in the cascade transformations exceed the corresponding $K_M$ values by several order of magnitude. A high ratio of substrate concentration/$K_M$ is, in general, beneficial for an efficient reaction into product. The DeoB is known to be somewhat vulnerable to inhibition by phosphate[79]. We determined the phosphate inhibition against Rib1P and showed it to be noncompetitive with a $K_i$ of 0.6 mM (Supplementary Fig. 2, Table 1). The inhibition by phosphate appears to be partial: DeoB retains about 10% of its basal activity in the absence of phosphate under the high-phosphate (≤ 1.0 M) conditions used for the cascade transformations (Table 1). Stability parameters were also obtained for the activity as well as for the protein structure (Table 1). UP is the most stable among the four enzymes used. It features a remarkable stabilization of structure by 1.0 M phosphate, reflected by 17 °C increase in melting temperature compared to 0.1 M phosphate (Table 1; Supplementary Fig. 3). YeiN and Yjjg exhibit melting temperature ($T_m$) in the range 36-46 °C and lose activity gradually (~50% in 16 h) even at 30 °C (Supplementary Figs. 3, 4). DeoB exhibits a $T_m$ larger considerably ($T_m \geq 11$ °C) than the $T_m$ of YeiN and Yjjg (Table 1). Its activity is however not as resistant as one might expect from the $T_m$. The DeoB is inactivated at a rate comparable to the inactivation rates of YeiN (Supplementary Fig. 4). Yjjg is the least stable enzyme.

**Three-step conversion of U into ΨMP**

The cascade reaction is shown in Fig. 3a. Offering **U** (244 g/L; 1.0 M) with phosphate present in 1.5-fold molar excess, the conversion into ΨMP proceeds to completion (≥95%), as shown in Fig. 3b. The evidence constitutes an excellent demonstration of the feasibility of the proposed three-enzyme cascade in Fig. 3a. The overall transformation happens in two kinetic phases, based on fast release of free Ura from **U** and slower consumption of uracil by β-C-ribosylation with Rib5P (Fig. 3b). Accumulation of the intermediary Ura to a concentration of ~450 mM is worth emphasizing because it reflects a temporary supersaturation of the reaction mixture in Ura, exceeding the Ura

solubility limit in water by ~20-fold (Supplementary Table 1). In the isolated YeiN reaction, ~25 mM Ura are maximally soluble (Supplementary Fig. 6). The curious delay in nucleation of the released Ura brings about a totally unanticipated feature of the cascade reaction that can have considerable advantage for the process engineering: tight monitoring of the enzyme activity ratio is not necessary to maintain control over the Ura solubility during the transformation. Were the formation of insoluble Ura happening at a rate comparable to the ΨMP formation rate, the overall reaction rate would have to be strictly limited by the rate of **U** conversion into Ura and Rib1P. Requirement to tune the flux through the individual steps of the domino reaction precisely would render even the basic reaction optimization a complicated task[80,81].

In Fig. 3b, given the specific activity of the mutase DeoB considerably lower than that of UP and YeiN, the isomerization of Rib1P to Rib5P was rate-limiting overall (Supplementary Fig. 7). DeoB is additionally inhibited by phosphate, as already mentioned (Table 1), plausibly explaining the ΨMP formation rate (~2 mM/min) about 30-fold lower than the nominal DeoB activity used in the reaction at 30 °C with 1.5 M phosphate.

Encouraged by these initial results, we systematically explored major reaction parameters with the aim of delineating the boundaries of the ΨMP synthesis. The ΨMP formation rate is largely independent of the pH in the range 6.5–7.5, but drops at pH 8.0 (Table 1, Supplementary Fig. 8a). It increases ~4-fold between 25 °C and 45 °C. At 50 °C, the ΨMP formation rate is dramatically decreased. Enzyme stability parameters (Table 1) would localize the upper limit of usage of YeiN and DeoB at ~40 °C, consistent with the results of conversion experiments showing an optimum ΨMP formation rate combined with full substrate consumption at 45 °C (Supplementary Fig. 8b). The $Mn^{2+}$ concentration, varied from $MnCl_2$ in the range 1.0 – 100 mM, is essential for DeoB and YeiN activity. Both enzymes are saturated with cofactor at $[Mn^{2+}] \geq 10$ mM, used in combination with [phosphate] $\geq 1.0$ M (Supplementary Fig. 8c). Use of too low $[Mn^{2+}]$ in the cascade reaction leads to excessive accumulation, and subsequent precipitation, of the released uracil. To avoid $Mn^{2+}$ precipitation as the phosphate salt, we limited the $[Mn^{2+}]$ to a maximum of 20 mM. Remarkably, the $Mn^{2+}$ precipitation does not affect the ΨMP formation rate at all, but we considered that insoluble material could complicate the product work-up (Supplementary Fig. 9). Reactions at variable [phosphate] (1.0–1.5 M) reveal that surplus phosphate is not necessary for the **U** conversion into ΨMP to complete. The ΨMP formation rate increases linearly with decreasing [phosphate], up to ~2.3-fold in the range used (Supplementary Fig. 8d, Table 1). Using reaction conditions updated suitably according to the results just described (40 °C, 10 mM $Mn^{2+}$, equimolar **U**, and phosphate), we increased the [**U**] in 0.2 M steps

**Table 1 | Summary of enzyme properties and characteristics of the enzymatic reactions**

| Enzyme[a] (EC number) | Expression[b] mg | Specific activity[c] U/mg, 30 °C (40 °C) | Specific activity under process conditions[d] 3-enzyme cascade U/mg | 4-enzyme cascade U/mg | $K_M$[e] mM | Cofactor/activator[f] | $\Delta G$[g] kJ/mol | Melting temperature[h] °C 1.0 M (0.1 M) $HPO_4^{2-}$ | Total turnover number of enzyme[i] 3-enzyme cascade (Fig. 3) | 4-enzyme cascade (Fig. 2) | pH range[j] |
|---|---|---|---|---|---|---|---|---|---|---|---|
| UP (2.4.2.3) | 150 | 56 ± 4.3 (80 ± 6.3) | 40 ± 1.4 | 12 ± 0.6 | 0.20 ± 0.01 (**U**), 2.3 ± 0.28 ($HPO_4^{2-}$)[107] | n.a./n.a. | 1.59 ($K = 0.54$)[107] | 82 (65) | $9.9 \times 10^6$ | $8.8 \times 10^7$ | 6.5–8.0 |
| DeoB (5.4.2.7) | 200 | 6.6 ± 0.1 (10 ± 0.7) | 0.6 ± 0.1 | 1.3 ± 0.1 | 0.06 ± 0.01 (Rib1P) $K_i$: 0.6 ± 0.1 ($HPO_4^{2-}$)[108] | $Mn^{2+}$, $Mg^{2+}$ / Rib1,5diP, Glc1,6diP | −8.42 ($K = 26$)[108] | 57 (65) | $2.4 \times 10^4$ | $1.5 \times 10^5$ | 6.5–8.0 |
| YeiN (4.2.1.70) | 100 | 2.5 ± 0.2 (3.8 ± 0.2) | 2.3 ± 0.2 | 1.6 ± 0.1 | 0.48 ± 0.1 (Rib5P), 0.59 ± 0.04 (Ura) | $Mn^{2+}$/n.a. | −24.12 ($K = 1.12 \times 10^4$)[73] | 42 (36) | $7.9 \times 10^4$ | $7.7 \times 10^4$ | 6.5–9.0 |
| Yjig (3.1.3.5) | 30 | 30 ± 2.0 (n.d.) | n.d. | 32 ± 0.5 | 3.4 ± 0.24 (**ΨMP**) | $Mn^{2+}$, $Mg^{2+}$/n.a. | −16.72 ($K = 6.4 \times 10^2$)[109] | n.d. (46) | n.d. | $8.2 \times 10^5$ | 6.5–8.0 |

*n.d.* not determined, *n.a.* not applicable.

[a] The abbreviations/names for the enzymes used are from Fig. 1.

[b] The amount of purified enzyme obtained from 1 L *E. coli* overexpression culture in shaken flasks is shown.

[c] The specific activities were measured with standard assays described in the Methods. For each enzyme, the concentration of the substrate(s) was saturating, exceeding the reported $K_M$ by at least 5-fold. One unit (U) is defined as the amount of enzyme converting one μmol of substrate/min under standard assay conditions. For UP, the activity is based on uracil released; for DeoB, the activity is based on Rib5P released; for YeiN, the activity is based on ΨMP released; and for Yjig, the activity is based on phosphate released.

[d] The (operational) specific activities were measured under conditions used in the 3-enzyme and 4-enzyme cascade reactions. For details, see the Methods section.

[e] Parameters are from initial-rate measurements in 50 mM HEPES buffer (pH 7.0) with 2.0 mM $MnCl_2$ at 30 °C. In two-substrate reactions of UP and YeiN, one substrate concentration was varied and the respective other was constant and saturating. The data are shown in Supplementary Fig. 1 together with non-linear fits of the rates to a single-substrate Michaelis-Menten equation. The inhibition constant $K_i$ is for an uncompetitive type of inhibition by phosphate. The inhibition data are shown in Supplementary Fig. 2.

[f] Cofactor and activator requirements are taken from the literature as follows. UP[107], DeoB[82], YeiN[69], Yjig[84,85].

[g] $\Delta G^{\theta} = -RT \ln K$; $K$ is the equilibrium constant based on literature data (pH 7.0); $T$ is temperature (311 K); $R$ is gas constant (8.314 J K$^{-1}$ mol$^{-1}$). The concentration of $H_2O$ in the YeiN reaction was taken as 55 M. The concentration of $Mg^{2+}$ or $Mn^{2+}$ was 1.0 mM. The $K$ values are from literature for YeiN[73], UP[107], and DeoB[79,108]. The $K$ value for the Yjig reaction with ΨMP and UMP as substrate was estimated using the online tool equilibrator[109].

[h] Melting temperatures determined from differential scanning fluorometry analysis at two concentrations of phosphate. The experimental melting curves are shown in Supplementary Fig. 3.

[i] Total turnover number (TTN) = $k_{cat}/k_d$. The $k_d$ is the apparent first-order inactivation rate constant estimated from inactivation time courses shown in Supplementary Fig. 4. The $k_{cat}$ is the apparent enzyme turnover number calculated from the specific activity measured under reaction conditions (see the relevant section of this table) prior to incubation[110]. The enzyme molarity is calculated from the protein concentration with the molar mass of the enzyme subunit. The values of $k_d$ are shown in detail in the legend to Supplementary Fig. 4 from one-pot transformation studies using the entire enzyme cascade (see Supplementary Fig. 8a).

[j] The pH range suitable for activity was inferred.

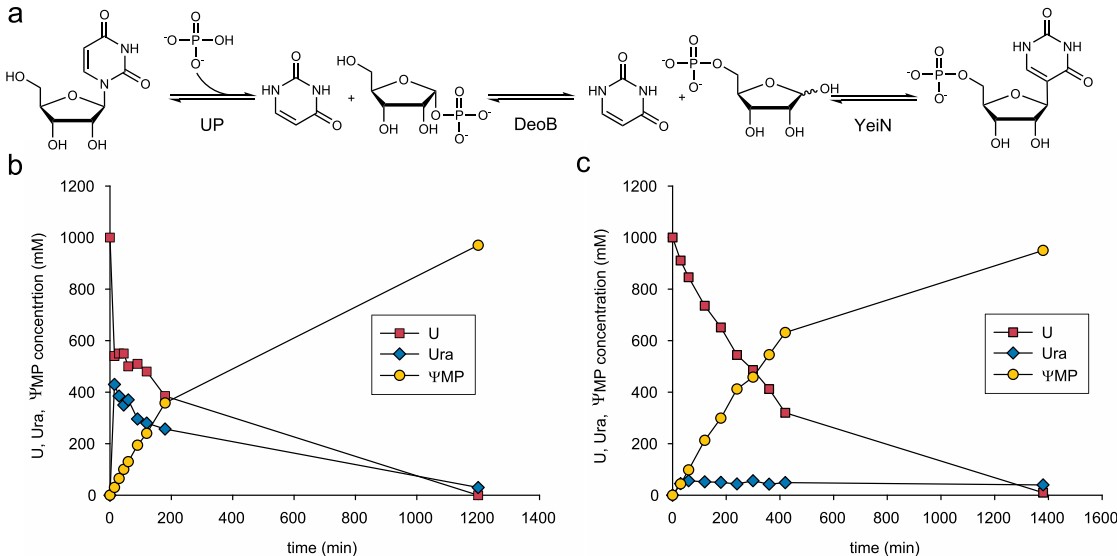

**Fig. 3 | Three-enzyme cascade synthesis of ΨMP. a** Scheme of the enzymatic cascade towards ΨMP. **b** Reaction time course of ΨMP synthesis at 100 μL (25 mg) scale. Here, 1.5 M potassium phosphate buffer pH 7.0, 1.0 M **U**, 10 mM MnCl₂, 0.1 mM Glc1,6diP, 1 mg/mL UP, 10 mg/mL DeoB and 6 mg/mL YeiN were incubated at 30 °C (n = 1 individual experiment). **c** Reaction time course of ΨMP synthesis at 5 mL (1 g) scale. Here, 1.0 M potassium phosphate buffer pH 7.0, 1.0 M **U**, 20 mM MnCl₂, 0.25 mg/mL UP, 2.5 mg/mL DeoB and 1.5 mg/mL YeiN were incubated at 40 °C (n = 1 individual experiment). Source data are provided as a Source Data file.

to a maximum of 2.0 M. The [ΨMP] released in 48 h increases with increasing [**U**], reaching an impressive ~1.7 M (550 g/L) at ~90% conversion of the **U** substrate (Supplementary Fig. 10). The ΨMP formation rate drops as the [**U**] increases, presumably due to effects of phosphate inhibition and fluid viscosity associated with the [**U**] change in combination. The marked loss in the ΨMP formation rate at 2.0 M **U** suggests an upper limit of cascade reaction efficiency reached at ~1.8 M substrate. A useful compromise between ΨMP formation rate (~5 mM/min) and final concentration (318 g/L; 98% conversion) is achieved at 1.0 M **U**. Keeping the enzyme ratio constant (UP:DeoB:YeiN = 1:10:6, by protein mass), the ΨMP formation rate depends linearly on the total protein concentration used (Supplementary Fig. 11). This provides a simple scaling factor of reaction engineering for facile process optimization towards conversion efficiency and cost-effectiveness.

Lastly, we consider the mechanistic requirement of DeoB to become activated via phosphorylation. The reversible conversion of Rib1P into Rib5P involves intramolecular phosphoryl transfer via a covalent DeoB phosphoenzyme intermediate. As in other sugar phosphate mutases, the activated DeoB state is that of enzyme phosphorylated at the active site nucleophile (Thr98 of DeoB). Sugar diphosphates, such as Rib-1,5-diphosphate (Rib1,5diP) and D-glucose-1,6-diphosphate (Glc1,6diP), serve as catalytic activators required in low micromolar concentration[82]. We show that the ΨMP formation rate is the same irrespective of whether Glc1,6diP (a costly reagent) is supplemented (Supplementary Fig. 12), indicating that activator addition is not necessary. A plausible explanation of the result is that the DeoB as-isolated is in the activated phosphoenzyme state. We note however that phosphoenzyme decomposition (e.g., by hydrolysis) could be a relevant factor of stability of the DeoB activity. Evidence of activity loss at temperatures of 20 °C below the $T_m$ suggests that inactivation might be caused by events different from thermal denaturation. At this stage of the development, however, enzyme stability was not an issue and its further exploration was left for consideration in the future.

Gram-scale preparative synthesis of ΨMP is performed in 5.0 mL volume, reflecting a scale-up factor of 50 compared to the reactions used in development. The reaction time courses obtained at the different scales are superimposable, as shown in Supplementary Fig. 13.

The excellent efficiency of the enzymatic transformation comes with the important benefit of a greatly simplified product isolation in technical-grade purity. Enzymes are removed by ultrafiltration, enabling their recycling, and the reaction mixture is lyophilized. The solid product recovered in ≥ 95% yield (based on UV-absorbance at 260 nm) is ΨMP of ≥97% HPLC purity (NMR, phosphate assay; Supplementary Figs. 14–20), with uracil, phosphate and potassium as minor impurities (~1% by weight each) and traces of Mn²⁺ present.

We made an effort to demonstrate the usability of the synthetic ΨMP in a follow-up transformation into a compound (ΨTP) involving more immediate applied interest than ΨMP itself. The ΨTP is substrate of in vitro transcription for mRNA synthesis. We use enzyme cascade phosphorylation of ΨMP (20 mM; 10 mL scale) according to protocol of our earlier study[73] (Supplementary Fig. 21a). The ΨMP conversion proceeds to 90% yield (Supplementary Fig. 21b), giving a final ΨTP concentration of 18 mM (9.7 g/L). The product is isolated by anion exchange chromatography followed by lyophilization to obtain a white crystalline ammonium salt of ΨTP (~100 mg). The chemical structure of the ΨTP is verified by ¹H, ¹³C, and ³¹P-NMR (Supplementary Figs. 22–24) and the purity is shown as ≥95% (Supplementary Fig. 25). These results validate the as-isolated ΨMP in technical-grade purity as fully sufficient for ΨTP synthesis.

### Discovery of Yjjg as a ΨMP-specific phosphatase

The three-step rearrangement of **U** appears to be well-suitable for ΨMP production. However, we figured that integration with existing chemical routes of RNA building block synthesis is better established for Ψ than ΨMP. Domino reaction in four steps is therefore designed to obtain Ψ from **U** (Fig. 2). In this reaction, phosphate serves to shuttle the Rib moiety and is recycled in the last step of the reaction. Proper function of the coupled enzyme system for the proposed synthesis relies on phosphate release specifically from ΨMP. A phosphatase well active with ΨMP and able to discriminate against Rib1P and Rib5P is not known. Commercial enzymes (e.g., calf intestine phosphatase) are not usable due to lack of specificity[83]. In a genome-wide study of *E. coli* phosphatases, Kuznetsova et al. analyzed the substrate spectrum of 23 enzymes of the structural superfamily of haloalkane dehalogenases. The phosphatase Yjjg (oher name: HAD5) exhibited fast turnover ($k_{cat} \geq 16$ s⁻¹) with pyrimidine monophosphates (UMP, dTMP, TMP)

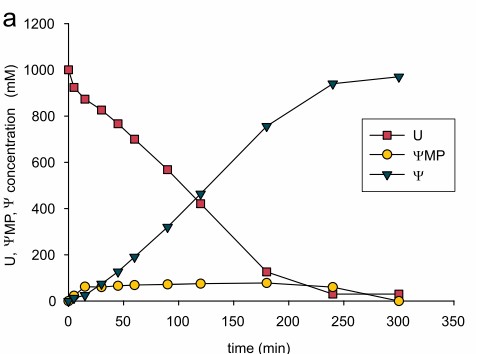
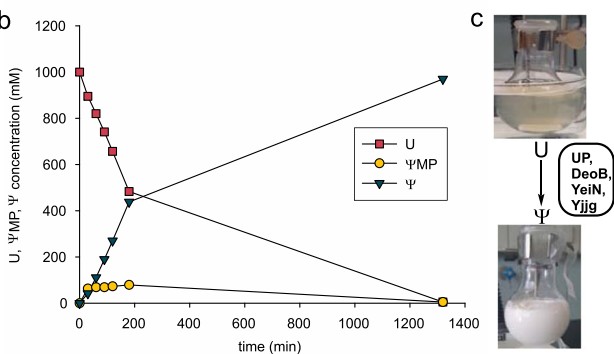

**Fig. 4 | Four-enzyme cascade synthesis of Ψ. a** Reaction time course of Ψ synthesis at 100 μL (25 mg) scale. Here, 0.1 M potassium phosphate buffer pH 7.0, 1.0 M U, 10 mM MnCl₂, 0.5 mg/mL UP, 5 mg/mL DeoB, 3 mg/mL YeiN and 0.2 mg/mL Yjjg were incubated at 30 °C (*n* = 1 individual experiment). **b** Reaction time course of Ψ synthesis at 100 mL (25 g) scale. Here, 0.1 M potassium phosphate buffer pH 7.0, 1.0 M U, 2.5 mM MnCl₂, 0.3 mg/mL UP, 2.5 mg/mL DeoB, 1.5 mg/mL YeiN and 0.2 mg/mL Yjjg were incubated at 30 °C (*n* = 1 individual experiment). **c** In situ crystallization of Ψ enables straightforward product isolation. Source data are provided as a Source Data file.

whereas no activity with Rib5P was detected[84–86]. We show here that in terms of catalytic efficiency ($k_{cat}/K_M$) the Yjjg prefers ΨMP (4.1 mM⁻¹s⁻¹) about 500-fold to Rib5P (0.008 mM⁻¹ s⁻¹) as a substrate for dephosphorylation (Supplementary Fig. 1f,g). This particular enzyme specificity thus renders Yjjg a clear candidate phosphatase to be used in the synthesis of Ψ from U.

### Four-step conversion of U into Ψ

We performed the cascade reaction as before for ΨMP production, with the exception that Yjjg is included and phosphate is added to just 10 mol% of the U substrate (four-enzyme cascade, Fig. 2). Pleasingly, as shown in Fig. 4a, the four-step reaction proceeds to completion (100% yield of Ψ) and the conversion rate is even faster (~5-fold) than in ΨMP synthesis. Lowered inhibition of the DeoB at decreased [phosphate] likely accounts for the observed rate acceleration. Interestingly, Ψ is released in the reaction to a concentration exceeding its solubility limit (200 mM) by ~5-fold (Supplementary Table 1). Indeed, once the rearrangement of U was complete, Ψ is observed to crystallize spontaneously from the reaction mixture. Analysis of the crystalline powder revealed pure Ψ (≥ 99%). The reader will notice that in situ crystallization is one of the most desirable approaches of reaction product isolation, especially at larger scale in process chemistry[87–89]. It is interesting that Ψ shows much lower solubility (~10-fold) than U. Preferred conformation in solution (Ψ *syn*; U *anti*)[90] might give an explanation. The anti conformation is expected to enable more extensive interactions with the water solvent than the *syn* conformation.

Contrary to ΨMP synthesis that involves phosphate as the substrate, the role of phosphate in Ψ synthesis is catalytic. The Ψ reaction thus required its own assessment of the [phosphate] to be used. Employing conditions otherwise as in Fig. 4a, the Ψ formation rate is shown to be dependent on the phosphate concentration (10–200 mM), with plateau (3.5 mM/min) reached at 100 mM (Supplementary Fig. 26a). The concentration of the intermediary ΨMP increases linearly with the [phosphate] used. At steady state during the reaction, the ΨMP accounts for ~80% of the total phosphate present (Supplementary Fig. 26b). The results imply limitation of the overall Ψ formation rate by the rate of dephosphorylation of ΨMP. Consequent variation of the Yjjg loading confirms the phosphatase as the rate-determining enzyme, with a maximum Ψ formation rate of 5.8 mM/min reached at 0.4 mg/mL (Supplementary Fig. 26a). Constancy of the Ψ formation rate at higher Yjjg loadings of up to 0.8 mg/mL suggests change in rate limitation to a different step, presumably that of DeoB. The portion of total phosphate bound in ΨMP reflects this change, by a

1.6-fold decrease to ~50% (Supplementary Fig. 27b). The lowered [phosphate] in the reaction also reduced the Mn²⁺ loading (2.5 mM) required to elicit the full enzyme activity (Supplementary Fig. 28a). Temperature variation reveals an optimum of the Ψ formation rate at 30 °C (Supplementary Fig. 28b), considerably lower than the temperature of maximum ΨMP formation rate (45 °C; Supplementary Fig. 8b). Sharp drop of the overall Ψ formation rate at elevated temperatures reflects slowdown of the conversion of ΨMP into Ψ and identifies Yjjg as the most thermolabile component of the enzyme cascade.

Using conditions updated for Ψ synthesis, we examined the U substrate concentration in the range 1.0–2.0 M. As already observed during ΨMP synthesis and even more strongly here, the Ψ formation rate decreases as the [U] is raised, suggesting a maximally usable substrate concentration of ~1.2 M (Supplementary Fig. 29). There may well be biological (inhibition) and physical limits (viscosity) of the Ψ formation rate under the conditions used. Considering that the product solution obtained from conversion of 1.0 M U already reflects a 5-fold degree of oversaturation in Ψ, further reaction intensification to increase the product concentration seemed unnecessary. Variation in total enzyme loading at constant enzyme ratio reveals a linear scaling of the Ψ formation rate with the protein concentration (Supplementary Fig. 30). Enzyme usage and time needed to fulfil a given conversion task are thus reciprocally interrelated process variables of key importance for optimization. Simple engineering relationships are evidently crucial in order to cope with the inherent complexity of multienzyme systems in the development of biocatalysis applications[42,80,91].

### Scale up of Ψ production

The scale-up involved a 1000-fold change in reaction volume from 100 μL in the characterization to 100 mL in the production. An intermediate step at 10 mL was used additionally. Agitation was changed from revolution in the tube to stirring with a magnetic stirrer bar in the flask. The enzyme loading was minimized towards the aim of full conversion of U in 24 h. As shown in Supplementary Fig. 31, the Ψ release up to a concentration of ~500 mM is identical within limits of error for the reactions done at different volume, indicating an entirely successful scale up. The reactions at 10 mL and 100 mL differ however from the 100 μL reaction in a much earlier onset of the crystallization of Ψ. Solid product starts to form already after ~3 h when the U conversion is at ~50%. Differences in vessel material (glass, plastic) and agitation mode used evidently result in strongly changed Ψ nucleation rates. However, the enzymatic rearrangement of the U proceeds

unaffected by the piling up of solid material and an identical amount of **Ψ** is obtained from the unit volume in all reactions. Once the **U** is exhausted, the liquid-solid suspension is cooled (−20 °C, 30 min) for the crystallization to complete. After lyophilization of the centrifuged solid, 24.3 g **Ψ** (white powder) is obtained in 95% isolated yield from the **U** used. The chemical structure of the **Ψ** is verified by $^1$H, $^{13}$C, and $^{31}$P-NMR (Supplementary Figs. 32–38) and the purity is shown as ≥99% by HPLC and assay for free phosphate.

## Discussion

Biocatalytic process technology for the production of **Ψ** is presented. Domino reaction in four enzyme-catalyzed steps achieves a highly efficient *N-C* rearrangement of **U**. The cascade transformation of **U** into **Ψ** deserves, in our view, the designation "dream reaction" from a general organic synthesis as well as process chemistry point of view[48,92–95]. It is atom-economic and selective in converting an expedient substrate (see below) in quantitative yield to the desired product. The reaction output exceeds by a large factor (≥ 5-fold) the thermodynamic boundaries of a stable product solution, enabling in situ crystallization of the **Ψ** for pure product isolation in ≥ 90% yield. The alternative (three-enzyme) cascade reaction to give **ΨMP** is similarly efficient by the criterions considered. Higher solubility of **ΨMP** than **Ψ** prohibits spontaneous crystallization. However, due to the excellent product purity (≥ 96%) received from the synthesis, a simple solvent removal is sufficient to isolate the **ΨMP**. Excluding steps required for the enzyme preparation, the production done at gram scale involves an extremely small E factor of just ~3 (**Ψ**) and ~1 (**ΨMP**)[38]. The waste is effectively water with phosphate, Mn$^{2+}$ and, unless recycled, enzymes remaining from the reaction. The process mass intensity (PMI) for the whole production (i.e., mass of all materials used, including the enzymes/mass **Ψ** or **ΨMP** isolated) is ~4.5. Small-molecule processes in the pharma industry typically involve PMI values of several 10 to well over 100[38]. Similarly, the E factors are ≥ ~25[38]. Compared to chemical syntheses of **Ψ**, the one-pot rearrangement of **U** into **Ψ** reported here offers fundamental improvements in process efficiency and sustainability, which both arise from the innovative cascade reaction development. Our considerations take into account recent developments in chemo-catalytic synthesis of *C*-nucleosides by List and co-workers[34], even though the application to **Ψ** has not been shown in their study. Besides the core features of the synthetic transformation already discussed, the absence of an isolated intermediate in the multistep reaction sequence, the avoidance of excess reagents or organic solvents used, and the consistent usage of simple process operations are important engineering characteristics of the biocatalytic process.

Reporting the chemo-enzymatic synthesis of the **U** derivative Molnupiravir, McIntosh et al.[50] at Merck questioned the use of **U** as substrate, arguing that it could not be considered a "true commodity raw material" by the requirements of sustainable and green production. The **U** would have to be synthesized in several chemical steps from Rib and uracil. This led to the suggestion of an alternative Molnupiravir route that circumvents **U** entirely and starts from Rib and uracil instead. Disregarding the difficult debate on supply chain risks, we suggest that **U** produced efficiently by microbial fermentation[75–77] must also be brought into the picture. Full techno-economic analysis and life cycle assessment of possible **Ψ** process options will eventually be required to decide between **U** or Rib (plus uracil) as substrate used in manufacturing. From a technical process chemistry viewpoint, however, the rearrangement of **U** presents an interesting alternative. In principle, however, the YeiN reaction could be integrated with enzymatic reactions for Rib5P formation from Rib[73]. Ribose phosphorylation by ATP is an established reaction at scale[50,51]. As an additional note on substrate used, our four-enzyme cascade is flexible to also work with **U**-5′-monophosphate (**UMP**). Depending on market situation, the **UMP** may present a cost advantage over **U**. As shown in Supplementary Fig. 39, release of phosphate from the **UMP** by Yjjg initiates the reaction sequence leading to *N-C* rearrangement into **ΨMP** and subsequent dephosphorylation to give **Ψ**. The **UMP** reaction differs from, and is less preferred from a sustainability point of view than, that of **U** in the larger (i.e., stoichiometric) amount of phosphate released into the product mixture.

Enzyme cascade transformations via phosphorylated or nucleotide-activated sugars enable innovative routes to complex molecules incorporating carbohydrate residue[41,96–100]. Applications are numerous in diversity-oriented synthesis at small scale[41,96,98,101,102] and interest for use in targeted production is rapidly growing[48,50,51,103]. However, telescoping multiple enzymatic reactions in one pot comes with significant challenges of process engineering (e.g., optimization) and control[38,42,99]. Here besides metrics of reaction efficiency already discussed, efficient use of enzyme catalyst is critical. Our g-scale synthesis of **Ψ** and **ΨMP** involves excellent protein mass-based turnover number (TONs) of 56 g/g and 129 g/g, respectively, received from just a single batch reaction without additional effort at enzyme recycling. On a mol basis, the individual enzyme TON is typically at ~$10^5$–$10^6$ (UP, Yjjg), somewhat smaller for YeiN and DeoB (~$10^4$) due to the enzyme's lower specific activity. The TONs calculated from the experimental conversion data approach in magnitude the TTN values (Table 1) estimated for the individual enzymes based on separately determined rate constants for reaction and inactivation. The implications are twofold. First, the cascade reaction performed in batch process succeeds in exploiting almost the full catalytic potential of the enzymes used (Yjjg, YeiN, DeoB), as given by their respective TTN. Second, further improvement of the TON will require enzyme stabilization, by immobilization for example, to enhance the TTN. Additional point of importance is that simple (linear) scaling relationships between the production rate and the total protein concentration will facilitate process optimization under the constraints from economic assessment. The current productivities, 36 g/L/h for **Ψ** (Fig. 4b) and 30 g/L/h for **ΨMP** (Fig. 3c) push boundaries in general cascade biocatalysis, demonstrating how truly efficient such multienzyme systems can be. The TTNs mentioned above and the phosphate inhibition of DeoB represent immediate targets for further development through integrative approaches of enzyme development and process engineering.

Lastly, the conversion of **U** into **Ψ** and **ΨMP** embodies new chemistry realized by cascade biocatalysis. To our knowledge, multistep enzymatic *N-C* rearrangement has never been shown in a relevantly similar form. In natural product glycosides, glycosyltransferase-catalyzed two-step *O-C* rearrangement via sugar nucleotide intermediate was suggested for flavonoid C-glucoside synthesis in our earlier work[104]. In providing synthesis routes of high efficiency, as shown here for **Ψ**, the concept of glycoside rearrangement of naturally abundant substrates promotes the quest of new biocatalytic cascades, potentially facilitated by emerging tools of computer-aided synthesis planning[102], and the search for the complementary set of enzyme activities.

## Methods
### Materials

Deuterium oxide (99.96%) was from Euriso-Top (Saint-Aubin Cedex, France). **U**, **UMP**, uracil, K$_2$HPO$_4$, KH$_2$PO$_4$, NaHCO$_3$, MnCl$_2$, HEPES, NaCl, IPTG, imidazole, tetra-n-butylammonium bromide, glycerol, and ATP were from Carl Roth (Karlsruhe, Germany). Glc1,6diP and SYPRO orange, was from Sigma Aldrich (St. Louis, USA). Rib5P was purchased from Biosynth (Staad, Switzerland). Expression vectors (pet15b or pet28a+) containing the genes for YeiN, DeoB, UP, and Yjjg were from Genescript (Leiden, The Netherlands).

### Enzyme preparation
N-terminally His$_6$-tagged YeiN, UP, DeoB, and Yjjg were used. All enzymes are encoded by *E. coli* genes. The relevant expression vector (pet15b _yein, pet15b _UP, pet15b_DeoB, and pet28a+_Yjjg) was

transformed into *E. coli* BL21_DE3 (pLys) and selected on 0.1 mg/mL ampicillin LB-agar plates. Yjjg transformants were selected on 0.05 mg/mL kanamycin LB-agar plates. Enzymes were expressed in 1-L baffled shaken flasks at 37 °C and 110 rpm, using 250 mL LB -media containing 0.1 mg/mL of ampicillin. For Yjjg, expression was done at 0.05 mg/mL kanamycin. Cultures were inoculated to an $OD_{600}$ of 0.1. At an $OD_{600}$ of about 0.8, the temperature was decreased to 18 °C and expression was induced with 0.4 mM IPTG (isopropyl β-D-1-thiogalactopyranoside) for 20 h. Cells were harvested by centrifugation at $4420 \times g$ at 4 °C for 30 min using a Sorvall RC-5B refrigerated superspeed centrifuge (Du Pont Instruments, Newtown, CT, USA). The supernatant was discarded and the pellet was suspended in His-tag binding buffer (50 mM HEPES, pH 8.0 containing 500 mM NaCl, 30 mM imidazole, 5% glycerol). Cells were disrupted using sonication (Fisherbrand Sonic Dismembrator, model Ultrasonic Processor FB-505; Fisher Scientific, Vienna, Austria; 8 min) on ice and the cell-free supernatant was recovered by centrifugation at 4 °C and $27,150 \times g$ for 50 min. Pre-treated cell lysate (20 mL) was loaded on $2 \times 5$ mL HisTrap FF column (Cytiva, Marlborough, MA, USA), equilibrated with His-tag binding buffer, and mounted on an ÄKTA prime plus (Cytiva) system. Protein purification was performed at 10 °C and flow rate of 3 mL/min. Protein was eluted using an imidazole gradient from 0% to 100% His-tag elution buffer (50 mM HEPES, pH 8.0, containing 500 mM NaCl and 300 mM imidazole). Fractions containing the target protein were pooled, concentrated, and buffer-exchanged with an Amicon Ultra-15 Centrifugal Filter Units (Millipore; Billerica, MA, USA). The final protein concentration was 30–150 mg/mL in 50 mM HEPES buffer containing 5% glycerol (v/v), 500 mM NaCl and 2 mM $MnCl_2$ (pH 7.0). Enzyme was stored at −20 °C until further use. Protein purification was monitored by sodium dodecyl sulfate–polyacrylamide gel electrophoresis (Supplementary Fig. 1).

### Initial rate analysis

This was used for standard enzyme activity measurement and determination of kinetic parameters. A 50 mM HEPES buffer (pH 7.0) containing 2.0 mM $MnCl_2$ was used at 30 °C. Standard activities were also recorded at 40 °C. Incubations were done in a total volume of 200 μL using an Eppendorf (Hamburg, Germany) thermomixer with gentle agitation (300 rpm). The substrate concentrations used are specified below for each enzyme. Reactions were started by adding concentrated enzyme solution (≤2% of total volume) to the temperature-equilibrated substrate solution. Activity determinations were performed at conditions of substrate saturation. Kinetic parameter determination involved initial rates measured at variable substrate concentrations (≥6 concentrations). In two-substrate reactions (UP, YeiN), the concentration of the second substrate was constant and saturating. Samples (5 μL) were taken at certain times and unless mentioned otherwise, quenched in methanol:water 1:1 (v:v; 100 μL), centrifuged, and analyzed by HPLC. Initial rates were acquired from the linear range of product formed (or substrate consumed) with time under conditions of ≤20% conversion of the substrate(s) used. One unit (U) is the enzyme amount for 1 μmol/min of product release or substrate consumed under the assay conditions used. Kinetic parameters ($V_{max}$, $K_M$) were obtained from non-linear least squares fit of Eq. (1) to the initial-rate data. Fitting was done with SigmaPlot V10.0 (Systat, Erkrath, Germany). In Eq. (1), $V$ is the initial rate dependent on [$S$], $V_{max}$ is the maximum initial rate (μM/min), $K_M$ is the Michaelis constant (mM) and [$S$] is the initial substrate concentration (mM). The specific activity was determined from the relationship $V_{spec} = V_{max}/[E]$, where [$E$] is the enzyme concentration. [$E$] was determined from the protein concentration, measured by 280 nm absorbance, and calculated with the protein-specific molar extinction coefficient.

$$V = \frac{V_{max}[S]}{(K_M + [S])} \tag{1}$$

The individual enzyme reactions were done as described below.

**YeiN.** The enzyme concentration was 0.025 mg/mL. Standard-specific activities were measured at 15 mM Ura and 5.0 mM Rib5P. For kinetic parameter determination, the Rib5P concentration was varied in the range 0.10–5.0 mM (10.0 mM uracil), the uracil concentration in the range 0.1–10 mM (5.0 mM Rib1P). Activities (U) and other rates refer to ΨMP released. Activity measurements were additionally performed under conditions mimicking those present during the enzyme cascade transformations. The activities are referred to as operational further on. Therefore, YeiN activity was recorded in 800 mM potassium phosphate buffer (pH 7.0) containing 20 mM $MnCl_2$, 15 mM Ura, 15 mM Rib5P, and 0.15 mg/mL YeiN at 40 °C (three-enzyme cascade). It was furthermore recorded in 20 mM potassium phosphate buffer (pH 7.0) containing 2.5 mM $MnCl_2$, 10 mM Ura, 1.0 mM Rib5P, and 0.15 mg/mL YeiN (four-enzyme cascade).

**UP.** The enzyme concentration was in the range 0.0012–0.011 mg/mL. Standard-specific activities were measured at 15 mM **U** and 40 mM phosphate. For kinetic parameter determination, the phosphate concentration was varied in the range 1.0–30 mM (20 mM **U**), the **U** concentration in the range 0.1–20 mM (30 mM phosphate). Activities (U) and other rates refer to uracil released. Operational UP activities were obtained in 1.00 M potassium phosphate buffer (pH 7.0) containing 20 mM $MnCl_2$, 1.00 M **U**, and 1.6 mg/mL UP at 40 °C (three-enzyme cascade) and in 100 mM potassium phosphate buffer (pH 7.0) containing 2.5 mM $MnCl_2$, 1.00 M **U** and 1.0 mg/mL UP at 30 °C (four-enzyme cascade).

**DeoB.** Enzyme-coupled assay was developed to measure the activity of DeoB. Initial rates were recorded in the presence of 10 mM uracil and 1.5 mg/mL YeiN. The DeoB was used at 0.0013 mg/mL. Under the conditions used, the DeoB activity is completely rate limiting and the ΨMP formation rate, therefore, equals the Rib1P isomerization rate. Standard-specific activity was measured at 3.0 mM Rib1P. For kinetic parameter determination, Rib1P was varied in the range 0.01–1.00 mM. To characterize the enzyme inhibition by phosphate, initial rates were measured at three different phosphate concentrations in the range 0.5–1.5 mM. The Rib1P concentration was varied in the range 0.03–0.6 mM. The inhibition was found to be noncompetitive, decreasing the $V_{max}$ while leaving the $K_M$ unaffected. The data were fitted with Eq. (2) where $K_I$ is the inhibition constant and [$I$] is the inhibitor concentration.

$$V = \frac{V_{max}[S]}{(K_M + [S])\left(1 + \frac{[I]}{K_I}\right)} \tag{2}$$

Activities (U) and other rates refer to ΨMP (=Rib5P) released. Operational DeoB activity in the three-enzyme cascade was obtained at 40 °C in 1.0 M potassium phosphate buffer (pH 7.0) containing 20 mM $MnCl_2$ and 1.00 M **U**. The reaction was started by adding 0.25 mg/mL UP, 1.5 mg/mL YeiN and DeoB in the strictly limiting concentration of 2.5 mg/mL. The operational activity in the four-enzyme cascade was obtained at 30 °C in 1.0 M potassium phosphate buffer (pH 7.0) containing 2.5 mM $MnCl_2$ and 1 M **U**. The reaction was started by adding 0.5 mg/mL UP, 1.5 mg/mL YeiN, 0.2 mg/mL Yjjg and DeoB in the strictly limiting concentration of 2.5 mg/mL. Retention of DeoB activity (~10% of basal activity) at the high phosphate concentrations used in the "operational" assays suggests that the inhibition by phosphate is only partial, with residual activity in the phosphate-saturated enzyme.

**Yjjg.** ΨMP or Rib5P was used as the substrate. The reaction was started by adding 0.0066 mg/mL (ΨMP) or 1.0 mg/mL (Rib5P) of enzyme. Samples (5 μL) were heat treated (95 °C, 5 min) and analyzed by HPLC (ΨMP reaction) or with a colorimetric phosphate assay[105] (Rib5P

reaction). Standard-specific activities were measured at 40 mM $\Psi$MP. For kinetic parameter determination, the concentration of $\Psi$MP or Rib5P has varied in the range 1.0–30 mM. Activities (U) and other rates refer to $\Psi$ or phosphate released.

## Thermal shift assay for melting temperature determination

Differential scanning fluorometry was used. Aliquots (45 μL) of enzyme solution (5 μM) in 0.10 M or 1.0 M potassium phosphate buffer (pH 7.0) were mixed with 5 μL of 200 × SYPRO orange solution and aliquoted in triplicate into a 96-well-PCR plate (Bio-Rad, Hercules, CA, USA). The plates were sealed with optical-quality sealing tape (Bio-Rad) and heated in a Bio-Rad CFX Connect Real-Time PCR Detection System, from 25 °C to 99 °C in increments of 0.5 °C and hold 30 sec at each step. The excitation and emission wavelengths were 490 and 575 nm, respectively, and the fluorescence changes in the wells were monitored continuously. CFX Maestro (Bio-Rad; Version 4.0.2325.0418) was used for data processing to calculate the melting temperature $T_m$. Buffer without enzyme was the control.

## Determination of nucleotide and protein concentration

Absorbance measurements were performed using a DS-11 Spectrophotometer (DeNovix, Wilmington, DE, USA). Uracil (258 nm, 8.3 mM$^{-1}$ cm$^{-1}$), **U** (262 nm, 10 mM$^{-1}$ cm$^{-1}$), $\Psi$ (262 nm, 7.5 mM$^{-1}$ cm$^{-1}$). Protein concentrations were measured at 280 nm and concentrations were calculated using the corresponding molar extinction coefficient and the molecular weight computed by protparam[106]. Yjjg (40,575 M$^{-1}$ cm$^{-1}$; 27,463 Da), YeiN (10,095 M$^{-1}$ cm$^{-1}$; 32,909 Da), DeoB (44,265 M$^{-1}$ cm$^{-1}$; 46,523 Da), UP (17,085 M$^{-1}$ cm$^{-1}$, 29,322 Da).

## High-performance liquid chromatography (HPLC)

Nucleosides and nucleotides were quantified by reversed-phase ion-pairing HPLC. Typically, 10 μL of diluted sample containing ~1 mM of analyte were loaded on a Kinetex C18 EVO column (Phenomenex, Aschaffenburg, Germany; 5 μm, 100 Å, 150 × 4.6 mm). Analytes were separated in 15-min long isocratic runs using 20 mM phosphate buffer, pH 5.9, containing 40 mM tetra-n-butylammonium bromide. The flow rate was 0.25 mL/min and the temperature set to 35 °C. Eluting compounds were detected at 260 nm. A 15-min HPLC trace is shown in Supplementary Fig. 40. Typical retention times were as follows: $\Psi$ (7.1 min), uracil (7.4 min), **U** (8.7 min), and $\Psi$MP (13.5 min).

To analyze the reaction mixture of conversion of $\Psi$MP to $\Psi$TP, the HPLC method was adapted slightly. A Kinetex C18 column (Phenomenex; 5 μm, 100 Å, 50 × 4.6 mm) was used. The eluting buffer from above additionally contained 12.5% (by volume) acetonitrile. A 5-min long isocratic run at 2.0 mL/min flow rate was used. Under these conditions, the retention times were as follows (Supplementary Fig. 41): Ura (0.3 min), $\Psi$MP (0.4 min), $\Psi$DP (1.1 min), ADP (1.1 min), $\Psi$TP (2.7 min), and ATP (4 min).

## Thin-layer chromatography (TLC)

About 1–2 μL sample were spotted onto a TLC plate (Merck, Darmstadt, Germany). The mobile phase was composed of 2-BuOH:AcOH:H$_2$O (2:1:1). The development solution consisted of 0.5 g thymol, 95 mL EtOH and 5 mL H$_2$SO$_4$. Spots were developed by heating the plate to -70 °C for 2 min using a heat gun.

## NMR analysis of purified reaction products

Lyophilized products were dissolved to 5.00–100 mM in D$_2$O in a total volume of 600 μL and transferred to a 5 mm high-precision NMR sample tube. Samples were analyzed on a Varian INOVA 500-MHz NMR spectrometer (Agilent Technologies, Santa Clara, California, USA) using the VNMRJ 2.2D software for the measurements, (Agilent Technologies, Santa Clara, CA, USA) or on a Bruker AVANCE III 300-MHz spectrometer (Bruker, Rheinstetten, Germany) with an autosampler and the Bruker Topspin 3.5 software for measurements.

## Reaction parameter variation for $\Psi$MP synthesis using the three-enzyme cascade

The standard (reference) conditions on which to perform single-parameter variation were 1.5 M potassium phosphate buffer (pH 7.0), 1.0 M U, 10 mM MnCl$_2$, 1.0 mg/mL UP, 10 mg/mL DeoB and 6.0 mg/mL YeiN. Reactions were performed in 100 μL scale and incubated at 30 °C. The pH was varied in the range 6.5–8.0 in 0.5 pH increments. [phosphate] was varied from 1.0 M to 1.5 M in 0.1 M increments. Temperature was varied in the range 30–50 °C in 5 °C increments. Addition of Glc1,6diP was tested at 100 μM. [MnCl$_2$] was varied in the range 1.0–100 mM, with no Mn$^{2+}$ addition as the reference. Variation in enzyme concentration was as follows: UP, 0.1–9.0 mg/mL; DeoB, 8.0–32 mg/mL; and YeiN, 3.0 – 12 mg/mL.

## Reaction parameter variation for $\Psi$ synthesis using the four-enzyme cascade

The standard (reference) conditions on which to perform single-parameter variation were 0.1 M potassium phosphate buffer (pH 7.0), 1.0 M U, 10 mM MnCl$_2$, 0.5 mg/mL UP, 5.0 mg/mL DeoB, 3.0 mg/mL YeiN and 0.4 mg/mL Yjjg. Reactions were performed in 100 μL scale and the standard temperature was 30 °C. Temperature variation was in the range 25–40 °C in 5 °C increments. [phosphate] was varied in the range of 10–250 mM. [MnCl$_2$] was varied in the range 1.0–10 mM, with no Mn$^{2+}$ addition as the reference. Yjjg was varied in the range 0.1–1.6 mg/mL.

## Four-enzyme cascade using UMP as the substrate

The $\Psi$ synthesis was performed in 0.1 M potassium phosphate buffer (pH 7.0), 1.0 M UMP, 10 mM MnCl$_2$, 0.5 mg/mL UP, 5.0 mg/mL DeoB, 3.0 mg/mL YeiN and 0.8 mg/mL Yjjg. Reaction were performed in 100 μL scale and incubated at 30 °C.

## Enzyme stability studies

Enzymes were incubated under conditions mimicking those of the cascade reactions to examine the stability of their activities. The conditions used were 1.0 M potassium phosphate buffer (pH 7.0) containing 20 mM MnCl$_2$ at 40 °C (three-enzyme cascade) and 0.1 M potassium phosphate buffer (pH 7.0) containing 2.5 mM MnCl$_2$ at 30 °C (four-enzyme cascade). The enzyme concentrations used in the incubations were as follows: UP, 0.3 mg/mL; DeoB, 4.0 mg/mL; YeiN, 2.1 mg/mL (three-enzyme cascade); UP, 0.3 mg/mL; DeoB, 4.0 mg/mL; YeiN, 3.0 mg/mL; Yjjg 0.1 mg/mL (four-enzyme cascade). Residual enzyme activity was measured in samples after 4 h and 16 h, using the enzyme-specific assays described above. Data were plotted in semi-logarithmic form (activity vs. time; Supplementary Fig. 4) and the approximate slope was used to estimate an apparent first-order inactivation rate constant ($k_d$). The total turnover number (TTN) of enzyme is given by Eq. (3) where $k_{cat}$ is the apparent catalytic constant which is calculated from the specific enzyme activity under these conditions (U/mg) and the enzyme molecular mass (g mol$^{-1}$). Note that the TTN thus calculated is based on mol.

$$\text{TTN} = \frac{k_{cat}}{k_d} \qquad (3)$$

## Synthesis of 1 g $\Psi$MP

The reaction was performed at 5 mL total volume in a 15 mL Sarstedt tube (Biedermannsdorf, Germany). Incubation was at 40 °C in a shaking water bath 1083 (GFL, Burgwedel, Germany). The conditions used were 1.0 M potassium phosphate buffer (pH 7.0) containing 20 mM MnCl$_2$ and 1.0 M U. The reaction was started with addition of 8 μL UP (150 mg/mL), 80 μL DeoB (161 mg/mL) and 60 μL YeiN (121 mg/mL), resulting in a final catalyst loading of 0.25 mg/mL UP, 2.5 mg/mL DeoB, and 1.5 mg/mL YeiN.

## Synthesis of Ψ in 1 g and 20 g scale

Reactions were performed at 10 mL or 100 mL total volume using a 250-mL round bottom flask placed in a water bath. The reaction mixture contained 1.0 M potassium phosphate buffer (pH 7.0) containing 2.5 mM $MnCl_2$ and 1.0 M **U**. The reaction was started with 0.25 mg/mL UP, 2.5 mg/mL DeoB, 1.5 mg/mL YeiN and 0.2 mg/mL Yjjg. For the 100 mL synthesis 0.20 mL UP (150 mg/mL), 2.50 mL DeoB (100 mg/mL), 1.25 mL YeiN (121 mg/mL and 1.43 mL Yjjg (14 mg/mL) were added to reach the desired biocatalyst loading. Incubation was at 30 °C and 900 rpm agitation rate. Temperature and stirring speed were controlled using a magnet stirrer with integrated heating plate (IK0003810000, Bitterfeld-Wolfen, Germany).

## ΨMP isolation

Enzymes were removed from 5 mL (∼1 M ΨMP) reaction mixture with an Amicon Ultra-15 Centrifugal Filter Unit (Millipore; Billerica, MA, USA) of 10 kDa molecular mass cut-off. The supernatant was lyophilized using a Christ Alpha 1-4 freeze drier (bbi-biotech GmbH, Berlin, Germany) attached to a Vacuubrand vaccum pump unit RZ 6. The solid recovered is ΨMP in technical-grade purity. The product was analyzed by HPLC and NMR, and phosphate was measured using a colorimetric assay[105]. The isolated yield reported is based on the mass calculated from the UV-absorbance of the isolated ΨMP. Prior to $^{31}$P-NMR measurements $Mn^{2+}$ was removed by treatment with Amberlite® IRC120 H.

## Ψ isolation

The reaction mixture from the Ψ synthesis (20 to 100 mL, ∼1 M Ψ) was cooled to −20 °C for 30 min to maximize Ψ precipitation. The supernatant was separated from precipitated Ψ using centrifugation at $2880 \times g$ and 4 °C for 10 min. The precipitated Ψ was lyophilized using instrument described above. The solid recovered is Ψ in high purity. The product was analyzed by HPLC and NMR.

## Water solubility of Ψ, U, and uracil

Solubility was examined in the range 30–50 °C in increments of 5 °C. The solid compound (Ψ, **U**, or uracil) was added in steps (1 mg) to Milli-Q water (1.0 mL) and incubated under agitation at 1200 rpm (Eppendorf thermomixer) for 30 min. Undissolved material was removed by centrifugation at $27,150 \times g$ for 5 min with an Eppendorf 5415 R microcentrifuge. The nucleoside and nucleobase concentration in solution was measured by 260 nm absorbance and calculated with the respective molar extinction coefficient, as described under "Determination of nucleotide and protein concentration".

## YeiN catalyzed conversion of 100 mM uracil

The reaction was performed in 1.0 mL total volume using a 2 mL glass vial placed in a water bath. Temperature and stirring were controlled using a magnet stirrer (IKA 3581200, Staufen, Germany) with integrated heating plate. The substrate solution contained 100 mM uracil and 150 mM Rib5P in water supplemented with 10 mM $MnCl_2$. The pH was set to pH 7.0 using 1 M NaOH. The reaction was started with 0.2 mg/mL YeiN and the incubation was continued at 37 °C and 900 rpm agitation rate. Samples were taken at certain times. They were centrifuged at 4 °C and $27,150 \times g$ for 1 min to remove insoluble uracil. The combined Ψ and uracil concentrations were determined photometrically, and the relative content was analyzed by HPLC, allowing for the determination of the absolute Ψ and uracil concentrations.

## Synthesis and isolation of ΨTP

The synthesis was performed in 10 mL total volume using a 15-mL Sarstedt tube for incubation at 37 °C in a water bath. The solution 50 mM HEPES buffer (pH 8.0) contained 20 mM ΨMP, 5.0 mM ATP, 60 mM phosphoenolpyruvate, 2.0 mM $MgCl_2$, 1.5 mg/mL cytosine 5′-monophosphate kinase[73] and 0.2 mg/mL pyruvate kinase (PK; Sigma-Aldrich, Darmstadt, Germany). The incubation was performed for 20 h. Suitable conversion of the ΨMP substrate (>90%) was shown by HPLC. Enzymes were removed with an Amicon Ultra-15 Centrifugal Filter Unit (Millipore; Billerica, MA, USA) of 10 kDa molecular mass cut-off. The filtrate was loaded onto a XK26/40 column packed with 125 mL DEAE FF anion exchange resin (both Cytiva) and mounted onto an ÄKTA prime plus FPLC system (GE Healthcare; Chicago, IL, USA). The column was equilibrated in 5 mM $NH_4HCO_3$ solution (pH 8.0). Elution was done at a flow rate of 10 mL/min using 200 mM $NH_4HCO_3$. Products were eluted at 80–100 mM $NH_4HCO_3$. Elution was monitored at 260 nm. Fractions containing ΨTP were pooled and concentrated using a Laborta 500-efficient rotary evaporator (Heidolph Instruments, Schwabach, Germany) operated at 40 °C and 30 mbar. The concentration was done until all $NH_4HCO_3$ was evaporated. Finally, the samples were lyophilized using a Christ Alpha 1–4 freeze drier (bbi-biotech GmbH, Berlin, Germany) using a Vacuubrand vacuum pump unit RZ 6. The isolated product was analyzed by HPLC as well as $^1$H, $^{13}$C, and $^{31}$P NMR (Supplementary Figs. 22–25).

## Synthesis and isolation of Rib1P

The synthesis was performed in 500 μL total volume in 1.4 M potassium phosphate buffer (pH 7.0) containing 2.0 M **U**. The reaction was started with 11 mg/mL UP and the incubation was continued at 40 °C and 350 rpm agitation rate overnight. Precipitated Ura was removed by centrifugation at 4 °C and $27,150 \times g$ for 10 min, followed by enzyme removal using an Amicon Ultra-15 Centrifugal Filter Unit (Millipore) with a molecular mass cut-off of 10 kDa. Phosphate was precipitated using barium acetate (mol equivalent of the phosphate present, used as crystalline powder) and the formed solids were removed by centrifugation at 4 °C and $27,150 \times g$ for 10 min. The supernatant was loaded onto a XK16/20 column packed with 25 mL DEAE FF anion exchange resin (both Cytiva) and mounted on an ÄKTA prime plus FPLC system (GE Healthcare). The column was equilibrated in 5 mM $NH_4HCO_3$ (pH 8.0). Rib1P was eluted using a linear gradient from 0 to 100% of 100 mM $NH_4HCO_3$ in 250 mL with a flow rate of 8 mL/min. Fractions were monitored for Rib1P (TLC) and phosphate content (colorimetric assay[105]). Fractions containing Rib1P and showing no detectable phosphate content were pooled and concentrated to a final volume of 50 mL using a Laborta 500-efficient rotary evaporator (Heidolph Instruments) operated at 40 °C and 30 mbar. The concentration was repeated as necessary until all $NH_4HCO_3$ was evaporated. The Rib1P concentration of the stock solution was determined from the phosphate released upon incubation with calf intestine phosphatase. The chemical structure and identity of the isolated compound as Rib1P were confirmed by $^1$H NMR (Supplementary Fig. 42).

## Reporting summary

Further information on research design is available in the Nature Portfolio Reporting Summary linked to this article.

## Data availability

All data reported in the paper are available from the corresponding author (B.N.) upon request. Additionally, source data are provided in this paper for Figs. 3,4, and Supplementary Figs. 1–13, 21, 26–31, and 39. Source data are provided in this paper.

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

## Acknowledgements

We thank Saskia Hofer (Institute of Biotechnology and Biochemical Engineering, Graz University of Technology) for expression and isolation of enzymes; and Prof. Hansjörg Weber (Institute of Organic Chemistry, Graz University of Technology) for NMR measurements. Funding by FFG (Austrian Research Promotion Agency) and COMET K2 program (acib: next generation bioproduction) awarded to B.N. is acknowledged.

## Author contributions

M.P. design of the study; experiments and data analysis; writing of the paper. A.R., experiments and data analysis, writing of the paper; B.N. design of the study; funding acquisition; discussion; writing of the paper.

## Competing interests

The authors declare no competing interests.
