## [Peer Review File · Nature Communications]

A selective and atom-economic rearrangement of uridine by cascade biocatalysis for production of pseudouridineREVIEWER COMMENTS

Reviewer #1 (Remarks to the Author):

The manuscript by Pfeiffer et al describes a biocatalytic cascade for production of pseudouridine, a key intermediate in the synthesis of pseudouridine triphosphate, a building block used in the manufacture of mRNA vaccines. The approach uses a uridine phosphorylase, phosphopentomutase, nucleotide monophosphate glycosidase and nucleotide phosphatase (YjjG) to convert uridine into pseudouridine. Although YjjG is known to catalyze the dephosphorylation of non-canonical nucleosides, the authors report for the first time that this enzyme displays activity towards pseudouridine but has no activity towards intermediates ribose-1-phosphate or ribose-5-phosphate. The authors scale the 4-enzyme cascade reaction to up to 25g and can achieve excellent productivity and isolated yields. Pseudouridine and N1-methyl pseudouridine are certainly important building blocks and more efficient methods for their production will be of broad interest.

In my opinion the last step of the cascade (dephosphorylation of the nucleotide monophosphate) is unnecessary as the ultimate target is the pseudouridine triphosphate. There is a lot of literature precedent for the phosphorylation of nucleosides using kinases, which proceed via the monophosphate to provide the nucleotide triphosphates. It also seems that during the scale up and isolation of the pseudouridine monophosphate, the phosphate salts were not removed meaning this product requires further purification (not performed).

Before publication, the following points should be addressed:

- 1) The abstract states 'current mRNA vaccines feature uniform substitution of uracil to pseudouracil in 1-N-methylated form'. Although the currently marketed COVID vaccines from Moderna and Pfizer contain N1-methylpseudouridine, not all vaccines under development do. Other modifications including pseudouridine and 5-methylcytidine are also used.
- 2) It is not clear what is meant by the terms 'Rib shuttle' or 'just-in-time release' used in the abstract. These terms should be fully explained.
- 3) Where the K_M values provided in Table 1 obtained experimentally by the authors? Kinetic experiments are not described in the methods section, and no Michaelis-Menten curves are shown in the SI. UP catalyzes the reversible reaction of uridine to uracil and ribose-1-phosphate, so it is not true that uracil and ribose-1-phosphate inhibit the enzyme.
- 4) Pg4 - Please clarify what is meant by 'The proton is not directly involved in any of the reaction'. Which proton is this referring to?
- 5) The authors refer to 'the ψ MP rate' which would be better described as the rate of ψ MP product formation. For example, 'DeoB is additionally inhibited by phosphate, plausibly explaining the ψ MP rate (~ 2 mM/min) about 10-fold lower than the nominal DeoB activity used in the reaction.'
- 6) Can the authors explain the role of glucose-1,6-bisphosphate in the DeoB reaction and provide references? Panoslan et al, J. Biol. Chem. 2011, 286, 8043 suggest that glucose-1,6-bisphosphate is not an essential cofactor but rather an activator. It phosphorylates the catalytic nucleophile which stimulates the reaction and is needed in $< 2.5 \mu\text{M}$
- 7) Pg 5 - Please explain what is meant by 'this evidence constitutes an excellent proof of the principle of figure 2, figure 3b'? It is not clear what principle they are referring to.
- 9) Pg6 – line 139-140 and 148 the authors states that DeoB is inhibited by phosphate, but also that DeoB exhibits full activity at phosphate concentrations exceeding 1M. Please clarify.
- 10) Pg7 line 181 'In this reaction, phosphate serves to shuttle the Rib moiety'. Please clarify what is meant here.
- 11) The authors describe a scale up of ψ MP to 1g and describe a product isolation. They state 'Higher solubility of ψ MP than ψ prohibits spontaneous crystallization. However, due to the excellent product purity ($\geq 96\%$) received from the synthesis, a simple solvent removal is sufficient to isolate the ψ MP'. How were the phosphate buffer salts removed? The methods section describes a lyophilization step which would not remove phosphates, so it is not clear how the product was achieved in 96% purity.

12) I do not agree with the claim that 'From a technical process chemistry viewpoint, however, the rearrangement of U is clearly preferred. Ribose phosphorylation by ATP is a complex reaction to be done at scale, in particular when involving an oxidase reaction (O₂ substrate, H₂O₂ product) to form the ultimate phosphate donor (acetylphosphate) in situ'. There are many examples of large-scale phosphorylation reactions employing ATP, such as the Merck Molnupiravir biosynthesis quoted in the manuscript. H₂O₂ by-product can be easily sequestered by catalase. Starting from ribose rather than uridine has many advantages, 1) ribose is cheaper and more sustainable, 2) ribose-5-phosphate can be generated in a single step from ribose using a ribokinase, which avoids the need for a phosphopentomutase. PPM is the rate limiting step in this synthesis, it is inhibited by phosphate, and high PPM concentrations are required (5 mg/ml purified DeoB compared to 3.6 mg/ml ribokinase used as clarified cell lysate - McIntosh et al ACS Cent. Sci. 2021, 7, 1980–1985)

13) The authors state 'A phosphatase active towards pseudouracil and able to discriminate against Ribose-1-phosphate and ribose-5-phosphate is unknown'. Although it has not been reported previously that YjjG accepts pseudouridine, it has been previously characterized by other groups as a promiscuous nucleotide phosphorylase. The following papers should be referenced M. Proudfoot et al J. Biol. Chem. 2004, 279, 54687; B. Titz et al, FEMS Microbiol. Lett. 2007, 270, 49

14) The optimal temperature for the four-step cascade process is lower than the three enzyme cascade process and the authors claim this is due to YjjG being thermolabile. Can the authors measure the melting temperature of the enzyme to confirm this? Could the difference in activity be attributed to the final product precipitating during the reaction at lower temperature?

15) The term PMI is not used correctly. PMI is the mass of product/mass of all materials. This includes the mass of the solvents used in the reaction and work-up steps, all buffer components and catalysts.

Reviewer #2 (Remarks to the Author):

The manuscript by Nidetzky research group titled "A perfectly atom-economic rearrangement of uridine by cascade bio-catalysis for production of pseudouridine" described an efficient strategy for synthesis of pseudouridine from uridine. The process takes advantage of an enzyme YeiN, a previously reported pseudouridine phosphorylase, which could catalyze reaction of the synthesis of pseudouridine or the hydrolysis of pseudouridine. The reversible reaction was broken by coupling the reaction with a phosphatase, which could hydrolyze pseudouridine-5-phosphate more efficiently. This is a very popular strategy in biosynthetic strategies to break the reversible reaction by coupling an irreversible reaction.

As pseudouridine is a very important starting material in mRNA drug area. This reviewer supports its publication in nature communication if the following issues can be addressed.

1. The authors claimed that pseudouridine can be precipitated during the reaction process. As the structure of pseudouridine is similar with uridine. Why uridine is soluble while pseudouridine can come out of the reaction system? Please explain.

2. In the manuscript, reactant solubility was discussed many times. Indeed, reactant solubility, especially uridine and pseudouridine, is one of the most issues to establish the current process. The authors claimed the data was shown in Table 1. However, Table 1 is only about enzyme properties. Please clarify.

3. The manuscript only discussed how much enzyme was added in reaction system. More details should be investigated. Enzyme stability at 37 degree (YeiN, YijG, UP, DeoB), pH range, enzyme activity with different metal ions. How about Mg²⁺?

4. The authors claimed that the synthesis is very efficient. However, I noticed that in the reaction system, the enzyme concentration is very high. Up to 2.5 mg/mL DeoB and 1.5 mg/mL YeiN were used in a 100 reaction system. This is a very high concentration. What concentration is the minimum usage for complete reaction? I think the data that shown in SI is not enough. Please explain. If the concentration of both enzyme was reduced, the reaction can't proceed cleanly. The incomplete reaction will increase the product purification difficulty.

5. Line 189, "whereas no activity with Rib5P was". a word is missed here.

Reviewer #3 (Remarks to the Author):

This is an outstanding and beautifully written paper describing a highly creative solution to an important problem. N1-Methylpseudouridine triphosphate is a necessary precursor to the highly effective mRNA vaccines and this paper describes an exceptionally facile enzymatic synthesis to its precursor, pseudouridine (and pseudouridine monophosphate) from readily available uridine. The key reaction in their enzymatic cascade, the formation of the C-C linkage between uracil and ribose catalyzed by the YeiN enzyme (running in reverse of its normal glycosylase reaction) was reported by this same group in a Nature Communications paper in 2020 (reference #68) . The key innovation described in this new paper is the combination of the correct enzymatic reactions together in one pot such that uridine can be used as the starting material for the synthesis of either pseudoU or pseudoUMP. Amazingly, these authors have devised a one pot synthesis where all the atoms of both the sugar and base in the final product pseudouridine are derived from uridine (maximum atom economy)! They describe the four enzyme, one pot combination of uridine phosphorylase to break the glycosidic linkage in uridine, DeoB to isomerize the ribose 1-phosphate product to ribose 5-phosphate, YeiN to form the new C-C bond in pseudoU-5-phosphate and the selective phosphatase Yjg to generate pseudouridine at concentrations where it precipitates from solution! Fantastic! Yields are terrific, purification is trivial. They also show that scale up to 100 mL reactions proceed as expected (even better!). Products and enzymes are rigorously characterized.

Response to reviewers/Revision

Please find below detailed response to the points raised in the review of our manuscript. We are grateful for the consideration of our manuscript and constructive criticism provided. Our responses are formatted in light blue color and are additionally marked with Response. We describe our position with respect to each point raised and explain the changes made upon revision. Overall, we believe to have addressed all points in an exhaustive manner.

REVIEWER COMMENTS

Response in general

We are grateful for a thorough and constructive review by the three Reviewers. More detailed responses are provided below. We agree, in general, with the idea that cascade reaction development benefits from a thorough understanding of the individual components of the whole reaction system. We have therefore performed an extensive kinetic characterization of each enzyme used and have further obtained additional parameters of enzyme stability.

However, it is important to clarify (and we will discuss this in more detail below) that in our opinion, the general notion of cascade transformation development by an approach of “dissection and reconstruction” is not the most efficient one. Dissection of system into the individual components to study them separately can be insightful, but it misses entirely the point about interactions. One can of course study each possible interaction in isolation, or in various combinations of factors, on the isolated enzyme components. But this procedure rapidly escalates into a huge number of experiments of unclear importance and renders the whole development completely inefficient. Plus, there is no guarantee that the evidence on isolated components provides a conclusive picture for system reconstruction. Therefore, what is really needed is a holistic approach that examines the components together in the way they are meant to be used, but succeeds at the same time in tracking the individual reactions through proper analytics. We have taken this holistic (integrative) approach from the beginning of this study. The conclusions from the approach are clear. And they are fully sufficient for systematic optimization of the multicomponent system. While we agree that the individual enzyme characterization done now during the revision has certainly added well to the overall body of evidence presented in the manuscript, there isn't really so much more that the enzyme parameters tell in conclusion that we haven't been able to conclude from the holistic approach already described in the original manuscript. We therefore consider the enzyme characterization, as it now stands after revision, to be comprehensive in relation to requirements of the current study.

We also address the issue of product purity. More detailed explanation is given below. For the vast majority of its possible applications, the pseudouridine 5'-monophosphate (Ψ MP) must be considered as a chemical intermediate. As far as we can see (in agreement with Reviewer 1), the most plausible follow-up transformation is synthesis of the triphosphate (Ψ TP) for use in in vitro transcription. Here we have shown by work done during revision that Ψ MP produced in the three-enzyme cascade is efficiently converted by enzymatic phosphorylation to the Ψ TP. No additional purification of the Ψ MP starting material is required except that enzymes are removed prior to phosphorylation. The residual phosphate from the reaction is so small (~1% by weight) that debate

about its removal is lacking basis in our opinion. Many phosphorylated chemicals supplied as reagents from commercial suppliers contain this or even larger amounts of phosphate. It is also not clear if there is any possible application of Ψ MP in which the remaining phosphate would really raise concern. We consider the evidence of excellent usability of the as-synthesized Ψ MP in the major follow-up transformation to be sufficient a demonstration of purity. There is no doubt that Ψ MP could be further purified to remove all of the phosphate, but purification at this stage has no clear significance or relevance for further use. We assume consensus on the notion that any research project requires prioritization, and the priorities are generally set based on rigorous assessment of importance.

Main changes: new data in Table 1, with associated text on page 6; experiment demonstrating the purity of Ψ MP for follow-up transformation on page 8; description of solubility on page 9; and consideration of amount of enzyme used in relation to the obtained turnover numbers on page 13. The fresh experimental evidence required the use of new methods. The Methods section is expanded to a considerable extent (e.g., initial rate analysis; differential scanning fluorometry for melting temperature determination; enzyme stability; phosphorylation of Ψ MP; synthesis of Rib1P required for kinetic analysis). The Supplementary Information was expanded by including the new results in full detail.

Reviewer #1 (Remarks to the Author):

The manuscript by Pfeiffer et al describes a biocatalytic cascade for production of pseudouridine, a key intermediate in the synthesis of pseudouridine triphosphate, a building block used in the manufacture of mRNA vaccines. The approach uses a uridine phosphorylase, phosphopentomutase, nucleotide monophosphate glycosidase and nucleotide phosphatase (YjjG) to convert uridine into pseudouridine. Although YjjG is known to catalyze the dephosphorylation of non-canonical nucleosides, the authors report for the first time that this enzyme displays activity towards pseudouridine but has no activity towards intermediates ribose-1-phosphate or ribose-5-phosphate. The authors scale the 4-enzyme cascade reaction to up to 25g and can achieve excellent productivity and isolated yields. Pseudouridine and N1-methyl pseudouridine are certainly important building blocks and more efficient methods for their production will be of broad interest.

Response: We thank the reviewer for the positive overall assessment of our study.

In my opinion the last step of the cascade (dephosphorylation of the nucleotide monophosphate) is unnecessary as the ultimate target is the pseudouridine triphosphate. There is a lot of literature precedent for the phosphorylation of nucleosides using kinases, which proceed via the monophosphate to provide the nucleotide triphosphates.

Response: We do not agree with the reviewer in the opinion that the in situ dephosphorylation step is unnecessary. Pseudouridine (Ψ) and Ψ MP are products each in its own right. Currently, most of the chemistry developed around the basic pseudouridine molecular structure is based on Ψ itself. In particular, synthetic methylation of N1 is reported only for Ψ . Chemical phosphorylation to prepare the triphosphate also starts from Ψ . Caution is suggested to believe that Ψ could be replaced just so by Ψ MP, without substantial adaptation of synthetic procedures. Consider the change in solubility properties imparted by the 5'-phosphate group. Synthetic transformations on Ψ often require

organic solvents. True, the preparation of the Ψ TP might ideally start from Ψ MP and bio-catalysis can be used for that purpose (see later). We have shown the synthesis of Ψ TP in an earlier Nature Communications paper and report here the use of the as-synthesized Ψ MP without any purification. The resulting Ψ TP is $\geq 95\%$ pure.

In summary, we retain the idea that both Ψ and Ψ MP deserve attention for synthesis by the biocatalytic cascade transformation. The manuscript conveys this position and hence we do not see the need for changes.

It also seems that during the scale up and isolation of the pseudouridine monophosphate, the phosphate salts were not removed meaning this product requires further purification (not performed).

Response: We already commented on this issue in the response to the Editor's request in the decision letter. The as-synthesized Ψ MP, recovered from the reaction by removal of the enzymes, contains roughly (actually it is a maximum) of 1 weight% of phosphate. Here, we wish to respond to the reviewer's point in two ways specifically. The first way is through the evidence obtained in the course of the revision, that the Ψ MP prepared as described in the manuscript is very well suited for follow-up transformation into the Ψ TP. We believe that enzymatic synthesis of Ψ TP might be the major route of synthetic use of the Ψ MP. The reviewer appears to have similar thoughts, judging from her/his comment above on the apparent lack of need for dephosphorylation of Ψ MP. The Ψ TP was prepared in very good purity of $\geq 95\%$ from the starting material in question. There is no evidence whatsoever that the small amounts of phosphate have any negative effect on the conversion into the Ψ TP. The amount of phosphate that may be carried into the product by way of commercial chemicals (e.g., ATP or others required in a phosphorylation cascade) could easily be higher considerably than the phosphate present in the Ψ MP. So, we conclude at this point that there is no reason to be concerned about the very minor phosphate content in the transformation to Ψ TP.

Secondly, we considered whether there might be additional applications of Ψ MP that could reasonably be compromised by the Ψ MP containing the 1% phosphate. Many phosphorylated chemicals offered commercially as reagents contain small amounts of phosphate. Considering the polarity of Ψ MP, water is a good solvent for the molecule. Enzymes work best in water whereas organic chemistry often requires organic solvents. For enzymatic reactions, phosphate is hardly ever a problem. Lastly, Ψ MP is not a pharmaceutical ingredient, as far as we know, so we do not consider molecule purity necessary for direct application as a drug substance. However, it is not immediately clear why minor phosphate impurity should be problematic for a pharmaceutical formulation of a drug substance that itself contains esterified phosphate. In summary, therefore, we could not think of a single application in which the residual phosphate would constitute a reasonable limitation. It certainly isn't one in the route to the Ψ TP as we have shown. Now, the phosphate in Ψ MP could be removed for sure (e.g., with chromatography certainly, electro dialysis could also be considered for application at large scale) but the main question is why, for which purpose.

The reviewer's comment would seem to us suggest a major point of criticism. It appears that the Editor has also understood the point as a major one. Based on response given above we wish to express disagreement with the reviewer's point in the stated implication that the Ψ MP was not pure enough and so required purification to be of further use. We took the comment of the reviewer into

very careful consideration, as shown by our efforts to demonstrate synthesis of the Ψ TP. However, one also has to say that in a scientific discourse any major doubt requires reasonable justification. Otherwise one is led inevitably into the unproductive realm of skepticism.

Evidently, a phosphorylated molecule obtained in water at neutral pH contains counter cations. We haven't considered removal of the cations by converting the Ψ MP into the acid form. The exact formulation of the product (e.g., potassium or sodium salt, acid) can be a point in specific applications not known at this moment. For enzymatic conversion of Ψ MP into Ψ TP, the cations (sodium in this case) are completely irrelevant. In our opinion, the residue of main concern might be Mn^{2+} from the reaction. In the phosphorylation considered for production of Ψ TP, the Mn^{2+} is excellently compatible because divalent metal ions (Mg^{2+} or Mn^{2+}) are required for the enzymes anyway. Again, removal of Mn^{2+} does not present a major technical difficulty but the purification must be done with specifications of the intended application in mind. To give an example, for ^{31}P NMR the Mn^{2+} ions (that would have interfered with the measurement) were removed efficiently by binding to ionic exchange resin.

Changes: We add the experiment to demonstrate conversion of Ψ MP into Ψ TP (page 8 – 9).

Before publication, the following points should be addressed:

1) The abstract states 'current mRNA vaccines feature uniform substitution of uracil to pseudouracil in 1-N-methylated form'. Although the currently marketed COVID vaccines from Moderna and Pfizer contain N1-methylpseudouridine, not all vaccines under development do. Other modifications including pseudouridine and 5-methylcytidine are also used.

Response: We respect the comment of the reviewer, but the abstract is a difficult place to consider these specific details. In our opinion, the statement used is not incorrect or biased. Vaccines under development may require privileged, often non-public information to know how their final structure will really look like. As far as we know, various mRNA vaccines were originally considered to feature uniform substitution of uridine by Ψ , now after further development the 1-N-methyl derivative is used.

Change: We alter the text to "currently marketed vaccines" which should satisfy the comment.

2) It is not clear what is meant by the terms 'Rib shuttle' or 'just-in-time release' used in the abstract. These terms should be fully explained.

Response/change: The sentence was changed for improved clarity. The sentence reads: "Coordinated function of the coupled enzymes in the overall rearrangement necessitates specific release of phosphate from the Ψ MP, but not from the intermediary ribose phosphates."

3) Where the K_M values provided in Table 1 obtained experimentally by the authors? Kinetic experiments are not described in the methods section, and no Michaelis-Menten curves are shown in the SI. UP catalyzes the reversible reaction of uridine to uracil and ribose-1-phosphate, so it is not true that uracil and ribose-1-phosphate inhibit the enzyme.

Response: The K_M values described in the original manuscript were from literature, partly based on our own work. In response to concerns raised in the review and per request of the Editor, we have

performed an extensive characterization of the enzymes used. Table 1 has been reworked to include the new material. The Methods have been updated to describe the assays used. The SI shows new results.

4) Pg4 - Please clarify what is meant by 'The proton is not directly involved in any of the reaction'. Which proton is this referring to?

Response: We wanted to say that none of the reactions of the cascades involved the proton in a sense of proton uptake or release during the reaction. Were this the case, the proton would be "directly" involved in the reaction as the substrate or the product. The implication of direct involvement of proton, in the sense meant, is a reaction equilibrium that is dependent on the pH (the proton concentration). We rewrote the sentence for clarity. The sentence reads: "The chemical reactions do not involve an immediate proton uptake or release."

5) The authors refer to 'the ψ MP rate' which would be better described as the rate **of ψ MP product formation**. For example, 'DeoB is additionally inhibited by phosphate, plausibly explaining the ψ MP rate (~ 2 mM/min) about 10-fold lower than the nominal DeoB activity used in the reaction.'

Response: We change to "formation rate" throughout.

6) Can the authors explain the role of glucose-1,6-bisphosphate in the DeoB reaction and provide references? Panoslan et al, J. Biol. Chem. 2011, 286, 8043 suggest that glucose-1,6-bisphosphate is not an essential cofactor but rather an activator. It phosphorylates the catalytic nucleophile which stimulates the reaction and is needed in $< 2.5 \mu\text{M}$

Response: We agree that the description of the sugar biphosphate as a "cofactor" was not precise and correctly stated, the term of an activator should be used. There can be two reasons, in our opinion, for why the extra addition of an activator, such as glucose 1,6-biphosphate, was not necessary for activity. One is that the enzyme as-isolated is already present in the activated form. We consider this explanation to be the most probable. The other is that during the reaction, perhaps due to the accumulation of Rib-1P and Rib-5P (this to a smaller extent than Rib-1P though), the enzyme can auto-activate itself by the release of small amounts of Rib-1,5-biphosphate. Loss of enzyme activity apparently somewhat faster than expected from the melting temperature might reflect gradual loss of the activated form. These explanations are speculative. However, stability of the DeoB is not a factor of the performance of the cascade reactions shown in the current manuscript. While interesting and potentially relevant when considering enzyme reuse, we considered a detailed analysis of DeoB stability to be beyond the scope of the current research.

Change: We added a short discussion to the revised manuscript (please see page 8; middle paragraph) and also included the suggested literature.

7) Pg 5 - Please explain what is meant by 'this evidence constitutes an excellent proof of the principle of figure 2, figure 3b'? It is not clear what principle they are referring to.

Response/change: The sentence was rephrased in an effort at clarification. The term "proof of principle" was used in the sense of realization of a certain method or idea to demonstrate the feasibility.

9) Pg6 – line 139-140 and 148 the authors states that DeoB is inhibited by phosphate, but also that DeoB exhibits full activity at phosphate concentrations exceeding 1M. Please clarify.

Response: We thank the reviewer for pointing out this error. As written, this is a contradiction. The DeoB is inhibited by phosphate, as we show with fresh evidence obtained in the course of the revision and summarized in the updated Table 1. The inhibition by phosphate is noncompetitive. At the conditions used in the cascade reaction, DeoB shows 5 – 10% of the activity of the fully active (uninhibited) enzyme. The native DeoB from *E. coli* is clearly usable, as shown, but the transformation would certainly benefit from a better enzyme. Improvement in enzyme performance might be achievable by testing a broader selection of mutases or by engineering. Development of an improved DeoB was not in the scope of this study.

Change: The original sentence at line 148 was changed.

Both enzymes are saturated with cofactor at $[Mn^{2+}] \geq 10$ mM, used in combination with [phosphate] ≥ 1.0 M (Supplementary Figure 3c).

10) Pg7 line 181 ‘In this reaction, phosphate serves to shuttle the Rib moiety’. Please clarify what is meant here.

Response: The unclear sentence was removed. The reference to Figure 2 seemed clear enough.

11) The authors describe a scale up of Ψ MP to 1g and describe a product isolation. They state ‘Higher solubility of Ψ MP than ψ prohibits spontaneous crystallization. However, due to the excellent product purity ($\geq 96\%$) received from the synthesis, a simple solvent removal is sufficient to isolate the Ψ MP’. How were the phosphate buffer salts removed? The methods section describes a lyophilization step which would not remove phosphates, so it is not clear how the product was achieved in 96% purity.

Response: We have responded already in some detail to this point above. Here we additionally show the determination of the product purity. After the synthesis, only 3 mol% (30 mM) of phosphate are not consumed. Together with 20 mmol/L of $MnCl_2$, a purity of 97% is calculated. The phosphate accounts for just 1 weight% of the product.

	$MnCl_2$	Ψ MP	Ura	phosphate
MW				
(g/mol)	126	324	112	95
final conc.				
(mM)	20	970	30	30
titer (g/L)	2.52	314	3.36	2.85
mass %				
(g/g)	1%	97%	1%	1%

As the reviewer suggested, the most likely use of Ψ MP is further phosphorylation by a kinase-cascade reaction to yield Ψ TP. To demonstrate the applicability of the follow-up transformation, we performed a small-scale reaction (10 mL, 20 mmol/L of Ψ MP). The reaction proceeded to 90%

conversion. By applying a single anion exchange chromatography using NH_4HCO_3 as eluent followed by lyophilization, we obtained ΨTP in a purity of 95% (HPLC) and in an isolated yield of >70%.

Changes: A new section in the Methods describes the associated methods. The results are included in the main text. Data are provided in the SI.

12) I do not agree with the claim that ‘From a technical process chemistry viewpoint, however, the rearrangement of U is clearly preferred. Ribose phosphorylation by ATP is a complex reaction to be done at scale, in particular when involving an oxidase reaction (O_2 substrate, H_2O_2 product) to form the ultimate phosphate donor (acetylphosphate) in situ’. There are many examples of large-scale phosphorylation reactions employing ATP, such as the Merck Molnupiravir biosynthesis quoted in the manuscript. H_2O_2 by-product can be easily sequestered by catalase. Starting from ribose rather than uridine has many advantages, 1) ribose is cheaper and more sustainable, 2) ribose-5-phosphate can be generated in a single step from ribose using a ribokinase, which avoids the need for a phosphopentomutase. PPM is the rate limiting step in this synthesis, it is inhibited by phosphate, and high PPM concentrations are required (5 mg/ml purified DeoB compared to 3.6 mg/ml ribokinase used as clarified cell lysate - McIntosh et al ACS Cent. Sci. 2021, 7, 1980–1985)

Response: We recognize and accept the points of the reviewer in a scientific debate. However, we do not share the opinion of the reviewer (see below). Our prime intention here is to make sure that our manuscript is not unnecessarily controversial. It may be sufficient to say in text that the route from Rib could be an interesting alternative. The revised manuscript has been updated in the way we believe it to be suitable.

Changes: The discussion on page 12 was shortened, with comments about ribose phosphorylation removed.

In the specific points of this debate, however, a detailed response may be expected from us. First of all, our discussion in the manuscript has included the statement (perhaps not seen by the reviewer) that we wished to leave aside the difficult questions about the substrate supply chain. We believe there is good reason to do so. Not only can markets be highly variable, but there are also numerous features of regional feature in the global market. Ribose is less costly than U or UMP, but the actual difference in price is not so clear. Uracil is needed additionally for the route from Rib. The claim that Rib is more sustainable than U or UMP cannot be verified from the reviewer’s statement. A detailed comparison of the two routes would be necessary and this is not available to our knowledge.

Concerning the issue of process chemistry, we probably should have said in our manuscript that “in our opinion the rearrangement of U is preferable.” Considering the comment of the reviewer, we realize that the wording was not good. It suggests clarity where in fact we only convey our opinion. However, the statements of the reviewer are opinion just as well and we do not see reason to share this opinion. The reaction cascade from pyruvate appears highly complex. It requires oxygen supply, probably by aeration (complexity factor #1), must manage hydrogen peroxide (complexity factor #2) and requires 3 enzymes to achieve a single reaction step (ATP supply for phosphorylation; complexity factor #3). We agree though that Rib phosphorylation from acetyl phosphate directly could be an interesting alternative. However, acetate is the stoichiometric byproduct, reaction monitoring for pH control is necessary. Phosphate would also be released in stoichiometric amounts when Ψ is produced.

It is clear from our analysis that the DeoB (mutase) step is limiting the overall productivity of the cascade reaction. However, our current study does not include search for the best suitable enzyme. We believe that the evidence of our study can serve as point of departure for other DeoB-like enzymes to be tested or existing enzymes to be engineered. This would follow the idea of “ideal process” and search of the “ideal catalyst” for that process. Just to say, the ribokinase mentioned by the reviewer is an engineered enzyme.

We are aware that our response is unlikely to convince the reviewer. The response is meant to emphasize the possibility of different points of view/opinions on the relevant issues. However, as we have pointed out, reconciliation of opinions should in no way be a criterion of acceptance of our manuscript.

13) The authors state ‘A phosphatase active towards pseudouracil and able to discriminate against Ribose-1-phosphate and ribose-5-phosphate is unknown’. Although it has not been reported previously that YjjG accepts pseudouridine, it has been previously characterized by other groups as a promiscuous nucleotide phosphorylase. The following papers should be referenced M. Proudfoot et al J. Biol. Chem. 2004, 279, 54687; B. Titz et al, FEMS Microbiol. Lett. 2007, 270, 49

Response/change: We agree with the reviewer and add the suggested references.

14) The optimal temperature for the four-step cascade process is lower than the three enzyme cascade process and the authors claim this is due to YjjG being thermolabile Can the authors measure the melting temperature of the enzyme to confirm this?

Response/change: We have determined melting temperatures in the reaction buffers and performed assays of enzyme stability. Results are shown in the updated Table 1 and in SI (Supplementary Figures 3). The new evidence confirms the conclusions.

Could the difference in activity be attributed to the final product precipitating during the reaction at lower temperature?

Response: The comment is unclear to us. Why would the activity of the enzyme affect the product solubility? After all, Ψ is the only product. Intermediates do not accumulate in significant amounts. At the 30°C used for the four-enzyme cascade transformation, all enzymes are sufficiently stable to perform the conversion unaffected by enzyme inactivation. Product precipitation appears to be an intrinsic property of the Ψ .

15) The term PMI is not used correctly. PMI is the mass of product/mass of all materials. This includes the mass of the solvents used in the reaction and work-up steps, all buffer components and catalysts.

Response: We agree with the reviewer and are grateful for the error pointed out. The values were recalculated.

Change: The process mass intensity (PMI) for the whole production (i.e., mass of all materials used: **U**, water, buffer, biocatalyst/mass Ψ or Ψ MP isolated) is ~4.5.

Reviewer #2 (Remarks to the Author):

The manuscript by Nidetzky research group titled “A perfectly atom-economic rearrangement of uridine by cascade bio-catalysis for production of pseudouridine” described an efficient strategy for synthesis of pseudouridine from uridine. The process take advantage an enzyme YeiN, a previously reported pseudouridine phosphorylase (please note: glycosidase), which could catalyze reaction of the synthesis of pseudouridine or the hydrolysis of pseudouridine. The reversible reaction was broken by coupling the reaction with a phosphatase, which could hydrolyze pseudouridine-5-phosphate more efficiently. This is a very popular strategy in biosynthetic strategies to break the reversible reaction by coupling an irreversible reaction.

As pseudouridine is a very important starting material in mRNA drug area. This reviewer support its publication in nature communication if the following issues can be addressed.

Response: We thank the reviewer for an overall favorable assessment of our research.

1. The authors claimed that pseudouridine can be precipitated during the reaction process. As the structure of pseudouridine is similar with uridine. Why uridine is soluble while pseudouridine can come out of the reaction system? Please explain.

Response: This is an interesting question. We only have a tentative answer. The polarity of glycoside linkage is different in the two molecules. The different linkages affect the preferred conformation in solution (see the figure below). The conformation in solution preferred in Ψ is *syn*, whereas the conformation preferred in U is *anti*. The *anti* conformation may enable more extensive interactions with the water solvent than the *syn* conformation. We considered DFT calculations but realized that this will easily expand into a new study and cannot be expected to yield unambiguous results reasonably quickly. Even though we cannot offer a molecular-mechanistic explanation better than informed speculation, we wish to emphasize that the difference in solubility is a plain fact supported by clear evidence. The claims we make are not based on interpretations of experimental findings, they derive from direct observations. We therefore feel that even though a precise molecular rationalization cannot be given, the evidence is clear enough in terms of its practical utility.

Change: We add a short statement (page 9) as tentative explanation of difference in solubility.

Neumann, J. M., Tran-Dinh, S., Bernassau, J. M. & Guéron, M. Comparative Conformations of Uridine and Pseudouridine and Their Derivatives. Eur. J. Biochem. 108, 457–463 (1980).

2. In the manuscript, reactant solubility was discussed many times. Indeed, reactant solubility, especially uridine and pseudouridine, is one of the most issue to establish the current process. The authors claimed the data was shown in Table 1. However, Table 1 is only about enzyme properties. Please clarify.

Response: We apologize for the error. There should have been a Supplementary Table S1 describing the solubilities. This has now been added.

3. The manuscript only discussed how much enzyme was added in reaction system. More details should be investigated. Enzyme stability at 37 degree (YeiN, YijG, UP, DeoB), pH range, enzyme activity with different metal ions. How about Mg²⁺?

Response: Table 1 was expanded with relevant characteristics of the individual enzymes. These characteristics include kinetic parameters, inhibition data and stability data. Activities and stabilities were also determined under operational conditions of the cascade reactions. Other factors (metal ion concentration, pH, reaction temperature) were analyzed in a holistic approach that applied all enzymes together in one pot. We commented on this approach already above in our responses to the Editor and Reviewer #1. We would like to emphasize that what we call holistic must not be mistaken with convoluted or lumped in a sense of limiting the interpretation. The comprehensive analytics applied has enabled us to relate effects of change of the bulk parameter on the activity of the individual enzyme. The enzymes YeiN, DeoB and Yjg require Mn²⁺ ion for activity. Mg²⁺ was not considered.

For the comment that it was only stated how much enzyme was added, please observe the extensive SI that shows the parameter variations used.

Change: Table 1 was updated with the enzyme characteristics mentioned. In the extent necessary, the characteristics are discussed in text. Methods have been expanded.

4. The authors claimed that the synthesis is very efficient. However, I noticed that in the reaction system, the enzyme concentration is very high. Up to 2.5 mg/mL DeoB and 1.5 mg/mL YeiN were used in a 100 mL reaction system. This is a very high concentration.

What concentration is the minimum usage for complete reaction?

Response: Our comment about “efficient reaction” was meant to primarily emphasize key metrics of conversion efficiency, like yield, final concentration and atom economy. We also meant productivity and enzyme turnover number, as discussed. It is true that the enzyme concentrations used are relatively high. However, the enzyme concentration as high or low is always relative to the product formed, hence the turnover number (TON) or total turnover number (TTN; a new parameter added during the revision). In these relevant metrics, the cascade reactions show excellent efficiency, as discussed. The productivity is a metric that scales directly with enzyme concentration. We have discussed this. In addition, we calculate the turnover number which normalized on enzyme mass. The turnover number is considered to be quite good when compared to other biocatalytic cascade transformations. In our opinion, this relativizes the use of high enzyme concentrations in the reactions. In the three enzyme cascade for synthesis of ΨMP, the minimum concentration of enzymes used to achieve full conversion was 0.25 mg/mL UP, 2.5 mg/mL DeoB and 1.5 mg/mL YeiN.

Change: Please see the discussion on page 12 – 13.

I think the data that shown in SI is not enough. Please explain. If the concentration of both enzymes was reduced, the reaction can't proceed cleanly.

Response: It is not completely clear from the comment of the reviewer which specific information is required as currently lacking. We have added Figure S7 which shows the enzyme concentration at which either YeiN or DeoB becomes rate-limiting in the cascade.

5. Line 189, "whereas no activity with Rib5P was ". a word is missed here.

Response: We updated the sentence. The word **detected** was added.

Reviewer #3 (Remarks to the Author):

This is an outstanding and beautifully written paper describing a highly creative solution to an important problem. N1-Methylpseudouridine triphosphate is a necessary precursor to the highly effective mRNA vaccines and this paper describes an exceptionally facile enzymatic synthesis to its precursor, pseudouridine (and pseudouridine monophosphate) from readily available uridine. The key reaction in their enzymatic cascade, the formation of the C-C linkage between uracil and ribose catalyzed by the YeiN enzyme (running in reverse of its normal glycosylase reaction) was reported by this same group in a Nature Communications paper in 2020 (reference #68). The key innovation described in this new paper is the combination of the correct enzymatic reactions together in one pot such that uridine can be used as the starting material for the synthesis of either pseudoU or pseudoUMP. Amazingly, these authors have devised a one pot synthesis where all the atoms of both the sugar and base in the final product pseudouridine are derived from uridine (maximum atom economy)! They describe the four enzyme, one pot combination of uridine phosphorylase to break the glycosidic linkage in uridine, DeoB to isomerize the ribose 1-phosphate product to ribose 5-phosphate, YeiN to form the new C-C bond in pseudoU-5-phosphate and the selective phosphatase Yjg to generate pseudouridine at concentrations where it precipitates from solution! Fantastic! Yields are terrific, purification is trivial. They also show that scale up to 100 mL reactions proceed as expected (even better!). Products and enzymes are rigorously characterized.

Response: We thank the reviewer for an overall favorable assessment of our research. No changes were necessary.

REVIEWERS' COMMENTS

Reviewer #1 (Remarks to the Author):

The authors have modified the text and completed a substantial number of additional experiments which have significantly strengthened the manuscript. In particular, I believe that the production of the ψ TP is an excellent addition. I enjoyed reading the paper and I am very supportive of publication.

Regarding my comments on the purity of the isolated ψ MP, I refer to the buffer salts and not only the free phosphate. Was the isolated yield determined by weight or UV? Either is fine but the authors should specify. The authors mention in their response that 'For enzymatic conversion of Ψ MP into Ψ TP, the cations (sodium in this case)...' but surely given the reactions are performed in potassium phosphate buffer the product would be a potassium salt? Even if all the phosphate were consumed in the reaction, ~60g/L potassium would remain. The authors should also consider the buffer salts added within the enzyme preparations. Three enzymes were prepared as solutions containing 50 mM HEPES buffer, 5% glycerol (v/v), 500 mM NaCl and 2 mM MnCl₂ (pH 7.0). Given the high final enzyme concentrations used in the reaction additional salts and glycerol have been added and these should also be considered in the calculation. From the experimental it is not possible to determine the volume added. The authors should add additional information to the experimental section to clarify, but I agree final product purification is not needed for this publication.

With regards to the kinetic characterization of the individual enzymes – While this is a nice addition to the paper, I agree with the authors that it is not entirely necessary when considering a cascade approach. What was unclear from the original submission was where the reported K_m values were obtained from as they were not clearly referenced.

Table 1: Units are defined in the materials section but it would be helpful to also define units in the table legend.

Reviewer #2 (Remarks to the Author):

My concerns have been addressed.

Another suggestion:

YjjG is not firstly discovered in attached manuscript. To acknowledge pioneer work made by other researchers, it is important to add some reference about this enzyme. For example, the work by Yakunin and co-workers (J Biol Chem 2015, 290 (30), 18678-98; J Biol Chem 2006, 281 (47), 36149-61) have investigated the substrate specificity of YjjG. They found YjjG could hydrolyze UMP but not sugar phosphate such as 6PGlu. I think this is an important clue for the establishment of the strategy in this work.

REVIEWERS' COMMENTS

Reviewer #1 (Remarks to the Author):

The authors have modified the text and completed a substantial number of additional experiments which have significantly strengthened the manuscript. In particular, I believe that the production of the Ψ TP is an excellent addition. I enjoyed reading the paper and I am very supportive of publication.

Response: We thank the reviewer for the positive evaluation of our revised manuscript.

Regarding my comments on the purity of the isolated Ψ MP, I refer to the buffer salts and not only the free phosphate. Was the isolated yield determined by weight or UV? Either is fine but the authors should specify.

Response: We thank the reviewer for the comment and agree that the method for yield determination needs to be clearly defined. The yield was based on the UV of the isolated Ψ MP. A defined sample of the powder was dissolved in buffer (pH 7.0) and the absorbance measured at 260 nm, using the mass of the isolated powder we were calculating the absolute amount of Ψ MP isolated and thus the corresponding yield.

Changes: We added a comment in brackets to the main text, stating that the yield is based on the absorbance of the isolated Ψ MP.

“The solid product recovered in $\geq 95\%$ yield (based on UV-absorbance at 260 nm) is Ψ MP of $\geq 97\%$ HPLC purity.” (p. 8)

Further change: The following sentence was added to the Methods:

“The isolated yield reported is based on the mass calculated from the UV-absorbance of the isolated Ψ MP.” (p. 18)

The authors mention in their response that ‘For enzymatic conversion of Ψ MP into Ψ TP, the cations (sodium in this case)...’ but surely given the reactions are performed in potassium phosphate buffer the product would be a potassium salt?

Response: Indeed, the product is a potassium salt. The statement in the Response Letter for the Revision was in error. We apologize for the confusion which is entirely on our part.

Even if all the phosphate were consumed in the reaction, $\sim 60\text{g/L}$ potassium would remain. The authors should also consider the buffer salts added within the enzyme preparations.

Three enzymes were prepared as solutions containing 50 mM HEPES buffer, 5% glycerol (v/v), 500 mM NaCl and 2 mM MnCl₂ (pH 7.0). Given the high final enzyme concentrations used in the reaction additional salts and glycerol have been added and these should also be considered in the calculation. From the experimental it is not possible to determine the volume added. The authors should add additional information to the experimental section to clarify, but I agree final product purification is not needed for this publication.

Response: To the 5 mL Ψ MP synthesis reaction in total 8 μL (150 mg/mL) UP, 80 μL (161 mg/mL) DeoB and 60 μL of YeiN (121 mg/mL) were added. Thus, only 150 μL of enzyme preparation were added, resulting in an addition of 5 mg NaCl and neglectable amounts of

glycerol, MnCl₂ and HEPES. Indeed, the counterion potassium would remain in the reaction, but as shown by the follow-up conversion to ΨTP this does not influence further use.

For isolation of ΨMP, barium precipitation followed by removal of barium using Na₂SO₄ could be an option. In our opinion, the technical grade purity is good enough for follow-up transformations and does not justify the generation of BaSO₄ waste.

Changes:

We updated the Methods section and give now the exact volume of enzyme preparation added to the synthesis.

For ΨMP synthesis:

“The reaction was started with addition of 8 μL UP (150 mg/mL), 80 μL DeoB (161 mg/mL) and 60 μL YeiN (121 mg/mL), resulting in a final catalyst loading of 0.25 mg/mL UP, 2.5 mg/mL DeoB, and 1.5 mg/mL YeiN.” (p. 17-18)

And Ψ synthesis:

“The reaction was started with 0.25 mg/mL UP, 2.5 mg/mL DeoB, 1.5 mg/mL YeiN and 0.2 mg/mL Yjgg. For the 100 mL synthesis 0.20 mL UP (150 mg/mL), 2.50 mL DeoB (100 mg/mL), 1.25 mL YeiN (121 mg/mL) and 1.43 mL Yjgg (14 mg/mL) were added to reach the desired biocatalyst loading.” (p. 18)

With regards to the kinetic characterization of the individual enzymes – While this is a nice addition to the paper, I agree with the authors that it is not entirely necessary when considering a cascade approach. What was unclear from the original submission was where the reported K_M values were obtained from as they were not clearly referenced.

Response: We are grateful for the clarification. The request for more detailed characterization of the enzyme kinetics was formulated by the Editor, based on the reports received for the original manuscript.

Table 1: Units are defined in the materials section but it would be helpful to also define units in the table legend.

Response: We thank the reviewer for the comment and added the Unit definition to the Table footnotes.

Changes: Table footnote

^c Specific activities were measured with standard assays described in the Methods. For each enzyme, the concentration of the substrate(s) was saturating, exceeding the reported K_M by at least 5-fold. One unit of activity (U) is defined as the amount of enzyme converting one μmol of substrate/min under standard assay conditions. For UP, the activity is based on uracil released, for DeoB the activity is based on Rib5P released, for YeiN the activity is based on ΨMP released and for Yjgg the activity is based on phosphate released.

Reviewer #2 (Remarks to the Author):

My concerns have been addressed.

Another suggestion:

YjjG is not firstly discovered in attached manuscript. To acknowledge pioneer work made by other researchers, it is important to add some reference about this enzyme. For example, the work by Yakunin and co-workers (J Biol Chem 2015, 290 (30), 18678-98; J Biol Chem 2006, 281 (47), 36149-61) have investigated the substrate specificity of YjjG. They found YjjG could hydrolyze UMP but not sugar phosphate such as 6PGlu. I think this is an important clue for the establishment of the strategy in this work.

Response: We thank the reviewer for the comment and acknowledged the important contribution of Kuznetsova et al (ref 90) to our research. Their work on HAD-like-phosphatases inspired our search to identify a specific phosphatase and we specifically mentioned it in our manuscript:

“In a genome-wide study of *E. coli* phosphatases, Kuznetsova et al analyzed the substrate spectrum of 23 enzymes of the structural superfamily of haloalkane dehalogenases. The phosphatase Yjjg (other name: HAD5) exhibited fast turnover ($k_{cat} \geq 16 \text{ s}^{-1}$) with pyrimidine monophosphates (UMP, dTMP, TMP) whereas no activity with Rib5P was detected^{82,83,90}.”

We also cited the work of Titz (ref 82) and Proudfoot (ref 83) for their detailed analysis of the substrate scope of Yjjg. We carefully considered the suggested reference of Yakunin et al., but in our opinion, it is less relevant as it focuses on HAD-like-phosphatases originating from *Saccharomyces cerevisiae*.